# U and Th content in magnetite and Al-spinel obtained by wet chemistry and laser ablation methods: implication for (U-Th)/He thermochronometer

Marianna Corre[1], Arnaud Agranier[2], Martine Lanson[1], Cécile Gautheron[3,1], Fabrice Brunet[1], Stéphane Schwartz[1]

[1]ISTerre, Univ. Grenoble Alpes, Univ. Savoie Mont Blanc, CNRS, IRD, Univ. Gustave Eiffel, 38000 Grenoble, France
[2]UBO, IUEM, Place Nicolas Copernic, 29820 Plouzané, France
[3]GEOPS, Univ. Paris-Sud, CNRS, Université Paris-Saclay, 91405 Orsay, France

*Correspondence to*: Marianna Corre (marianna.corre@univ-grenoble-alpes.fr)

**Abstract.** Magnetite and spinel thermochronological (U-Th)/He dates often display significantly dispersed values. In the present study, we investigated the contribution of analytical (and standardization) errors to this dispersion. U and Th content of magnetite (natural and synthetic) and natural Al-spinel samples with U and Th concentrations between 0.02 and 116 µg/g were analyzed using both wet chemistry and in-situ laser ablation sampling methods. New magnetite reference samples

(NMA and NMB) were synthesized consisting in U- and Th-doped nanomagnetite powders, whose U and Th concentrations were determined using wet chemistry method (U and Th of NMA is ~40 µg/g and NMB ~0.1 µg/g). We show that for both U and Th analyses, the reproducibility obtained with the wet chemistry protocol depends on the U and Th concentration. It is below 11% for U-Th values higher than 0.4 µg/g and reaches 22% for U-Th content lower than 0.1 µg/g. This result implies that (U-Th)/He thermochronological ages cannot be more reproducible than 24% for magnetite containing less than 0.1 µg/g

of U and Th, thus explaining part of the natural ages variability. U and Th data obtained by laser ablation ICP-MS on natural magnetite and Al-spinel samples were calibrated using both silicate glass standards and synthetic magnetite samples. The U and Th contents determined using NMA are consistent with those obtained by wet chemistry method but they are overestimated by 30% when using the glass standard samples only. These results highlight the impact of matrix effect on the determination of the U-Th content in magnetite. We thus recommend to use a well-characterized magnetite reference for the

calibration of the U-Th signals obtained by laser ablation. The scatter in the (U-Th)/He magnetite ages can be expected to be ~20% if the U and Th contents are determined by laser ablation. This level of precision is actually not significantly different from that obtained using wet chemistry method, which paves the way for the use of laser ablation for determining (U-Th)/He ages. In the absence of spinel reference for U and Th calibration using LA-ICP-MS, silicate glass references along with NMA were used. U and Th contents were found to be ~30% lower than the values obtained using wet chemistry. This

discrepancy underlines the importance of using a standard with a composition close to that of the mineral of interest.

Although magnetite and Al-spinel have related crystal-structures, the magnetite standard is not appropriate for U and Th analysis in Al-spinel using LA-ICP-MS.

**Key words:** (U-Th)/He, magnetite, Al-spinel, wet chemistry, laser ablation, mafic and ultramafic rocks

## 1 Introduction

In the last 15 years, the development of thermochronological (U-Th)/He methods applied to both magnetite (MgHe) and spinel (SpHe), has opened up new avenues for dating the exhumation of mafic and ultramafic rocks (e.g., Blackburn et al., 2007; Cooperdock and Stockli, 2016; Cooperdock and Stockli, 2018; Schwartz et al., 2020) and for the chronology of aqueous fluid - ultramafic rock interactions that produce magnetite through "serpentinization" reactions (e.g., Cooperdock et al., 2020; Cooperdock and Ault, 2020). Ultramafic rocks are widely exposed in orogenic and ophiolitic contexts on continents as well as at slow spreading centers, i.e., a couple of thousands of meters under the ocean. Accurate thermochronological data on ultramafic systems are thus essential to quantify the timing of the exhumation of mantle rocks in various geodynamic settings, like transform faulting in oceanic core complexes or upon the emplacement of ophiolitic units.

Both magnetite, $Fe_3O_4$, and spinel *ss*, $(Mg,Fe)(Al,Cr,Fe)_2O_4$, are iron-bearing oxides which crystallize in the spinel structure and often incorporate trace amounts of U, Th and Sm (at the ng/g levels) during their crystallization. Helium is barely soluble in minerals (Gautheron and Zeitler, 2020), and only radioactive He produced during alpha decay of radiogenic isotopes contained in the mineral structure or from neighboring minerals can accumulate in the crystal structure (e.g., Gautheron et al., 2022). The (U-Th)/He date acquisition requires the measurement of both radiogenic $^4$He on one hand and radioactive $^{235}$U, $^{238}$U, $^{232}$Th and $^{147}$Sm concentrations on the other hand. MgHe and SpHe dates obtained for a variety of geological cases display quite dispersed values, typically in the 5 to 70% range. Such variability could be explained by heterogeneous crystallization timing, variable He diffusion coefficient in those minerals, alpha-implantation from neighboring U-Th-rich mineral or associated with the very low U, Th and Sm content in those minerals (e.g., Cooperdock and Stockli, 2016; Copperdock and Stockli, 2018; Schwartz et al., 2020; Gautheron et al., 2022), which is therefore difficult to measure precisely. In addition, well-established magnetite and spinel samples as well as a collection of systematic analyses of U and Th precision and error for samples having different U and Th concentrations, are lacking for the generalization of this emerging dating method. Precision in Sm content determination will not be discussed here because data were not acquired in this study, as Sm is routinely analyzed.

In this contribution, in order evaluate the magnitude of analytical dispersion, we analyzed U and Th in samples of various origins, e.g., natural and U-Th doped home-made synthetic magnetite as well as natural Al-spinel, which display a wide range of U and Th concentrations. Firstly, a wet chemistry analysis of U and Th with isotopic dilution using an Inductively Coupled Plasma Mass Spectrometer (ICP-MS) has been performed. The latter is similar to the one proposed Blackburn et al.

(2007) and Cooperdock and Stockli (2016). In parallel, we explored in-situ quantification of U and Th concentrations in magnetite and Al-spinel by Laser Ablation (LA) sampling coupled to Inductively Coupled Plasma Mass spectrometer (LA-ICP-MS). The latter method, however, requires the use of appropriate solid standards with matrix effects similar to those

affecting the samples of interest (e.g., Steenstra et al., 2019; Koch et al., 2002). Matrix effect in the case of spinel (Cr-spinel) was only investigated so far for elements with much higher concentration (> 10 µg/g; Locmalis et al., 2011 and Colas et al., 2014) than typical U and Th concentration encountered in spinel (< 0,5 µg/g). The LA-ICP-MS results obtained here were calibrated using silicate glass standards, and also by using, in addition, a home-made synthetic U-Th magnetite. Reproducibility, accuracy, and applicability of both methods are discussed along with their impact on the determination of

(U-Th)/He dates.

## 2 Methods

### 2.1 Sample description and characterization

#### 2.1.1 Natural sample

Three natural samples were selected for this study (Fig. 1, Table 1), a magnetite-bearing standard (IF-G), magnetite single-

crystals (RB) from an alpine ophiolite and a purchased aluminous spinel crystal. Each sample was grounded in order to get homogeneous powders and XRD data were collected using a reflective geometry on Bruker diffractometer D8 Advance at ISTerre (France). The IF-G sample is a mixture of magnetite, quartz and actinolite (Govindaraju, 1995) sampled from a large iron-ore deposit of the Issua supracrustal belt, West Greenland, that is used as a standard for major elements (Govindaraju, 1995). IF-G mineralogy is confirmed here by the XRD data (Fig. S1). Reported U and Th concentrations in this sample from

9 independent studies range from 0.01 to 0.03 µg/g and 0.03 to 0.1 µg/g respectively (Govindaraju, 1995; Dulski, 2001; Bolhar et al., 2004; Kamber et al., 2004; Guilmette et al., 2009; Parks, 2014; Bolhar et al., 2015; Viehnmann et al., 2016; Table 1). The second sample (RB) is made of millimeter-sized euhedral magnetite crystals sampled in the Rocher Blanc ophiolite (western Alps, France; Tricart and Schwartz, 2006). Inclusion-free single-crystals of RB magnetite were already studied for MgHe thermochronology by Schwartz et al. (2020). Two grams of RB magnetite were grinded for the present

study. The U and Th contents of the RB euhedral magnetite are in the 0.006 to 0.029 µg/g range (Table 1). Finally, 2 grams of powder were obtained by grinding a 5-cm large crystal of aluminous spinel (Al-Spl) of unknown origin purchased in a jewelry store and selected because of its size and purity (Table 1). Al-Spl was analyzed by EDS SDD (Silicon Drift Detector) SAMx under a Scanning Electron Microscope (SEM) Vega3 Tescan at the ISTerre (France) and yielded a $(Mg_{0.65}Fe_{0.35})Al_2O_4$ composition.

## 2.1.2 U-Th doped synthetic magnetite samples

Two batches of U-Th doped nanomagnetite powder (named NMA and NMB) were synthesized by co-precipitation from FeCl$_2$ and FeCl$_3$ solutions (Martínez-Mera et al., 2007). Equal amounts of U and Th, ~40 µg/g for NMA and ~0.05 µg/g for NMB were added to the acidified FeCl$_2$ starting solution. All solutions were prepared using boiled de-ionized water (MilliQ, 18.2 MOhms), deoxygenated by bubbling with N$_2$ gas for 30 min. Instantaneous precipitation of magnetite was achieved at 45 °C by simultaneous addition of the 0.125 mol/L of FeCl$_2$ and FeCl$_3$ solutions to an ammonia solution at 0.2 mol/L. The solid was then separated from the supernatant using a permanent magnet and it was rinsed four times with oxygen-free MilliQ water to avoid oxidation. X-ray powder diffraction data diffraction obtained on the NMA sample indicated the production of 85% of magnetite and 15% of goethite and allowed to estimate the grain size to 15 nm from the diffraction peaks width (see Fig. S4 in the supplementary data). The two mineral phases could not be separated. Complementary experiments were performed to determine the U and Th host phase(s) in the nanomagnetite product, which are presented in Appendix A.

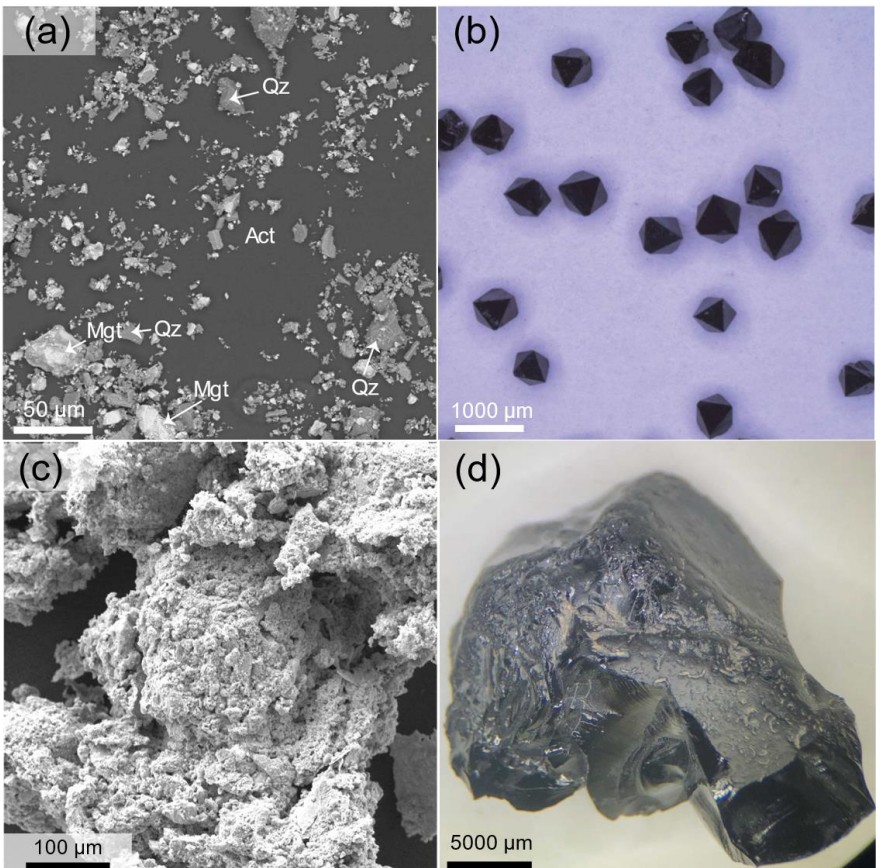

**Figure 1**. Images of the studied samples before the grinding. (a): BSE image of IF-G; (b): Photograph of magnetite crystals from Rocher Blanc (Alps, France); (c): BSE image of nanomagnetite particles (NMA sample) and (d): Photograph of the fragment of Al_Spl macrocrystal used for this study.

**Table 1.** Samples description.

| Sample | Issua magnetite-quartz (IF-G) | Rocher Blanc Magnetite (RB) | Aluminous spinel (Al_Spl) | Nanomagnetite powder A (NMA) | Nanomagnetite powder B (NMB) |
|---|---|---|---|---|---|
| Origin | Issua belt (Greenland) | Rocher Blanc ophiolitic massif (French Alps) | Unknown | Synthetic | Synthetic |
| Physical characteristics | Powder of 50 µm (53 vol.% of quartz, 37 vol.% of magnetite and 10 vol.% of actinolite). | Euhedral and pseudo-euhedral single crystals (400 to 600 µm across), containing inclusions of titanite, chlorite ilmenite and rutile | Single crystal (5 cm) | 15 nm powder (85 vol.% magnetite and 15 vol.% of goethite) | 15 nm powder (85 vol.% magnetite and 15 vol.% of goethite) |
| Chemical characteristics | U (µg/g): 0.013 to 0.03 Th (µg/g): 0.03 to 0.1 µg/g | U (µg/g): 0.008 to 0.029 Th (µg/g): 0.006 to 0.020; Schwartz et al. (2020) | U, Th, Sm (µg/g): unknow | U expected concentration: 40 µg/g Th expected concentration: 40 µg/g | U expected concentration: 0.05 µg/g Th expected concentration: 0.05 µg/g |
| Preparation for LA-ICP-MS analysis | Due to the mineral heterogeneity even after grinding IF-G was not analyzed by LA-ICP-MS | Grinding with a planetary mill in a 150 mL agate bowl with 10 agate beads of 10 mm diameter, 2 grams of magnetite and 100 mL ethanol for 2×10 min at 500 rpm 40 mg of powder is pressed at ~1000 MPa (20000 N) to have pellet with a diameter of 5 mm. Pellets are embedded in epoxy | 40 mg of powder is pressed at ~1000 Mpa (20000 N) to have pellet with a diameter of 5mm. Pellets are embedded in epoxy | | |

## 2.2 Sample preparation for U and Th analysis

To compare the U and Th concentrations obtained with wet chemistry and laser ablation sampling methods, 2 grams of natural samples (RB and Al_Spl) were homogenized. The studied materials were ground separately in a planetary mill at 500 rpm for 10 min to obtain a grain size of ~2 microns. Each sample was characterized by XRD (Fig. S1 to S4). The RB powder contains calcite, titanite, chlorite (Fig. S2), whereas Al_Spl sample is composed of pure aluminous spinel. Samples NMA and NMB which already consisted in nanoparticles did not necessitate grinding. The U and Th contents of these 4 samples were retrieved using wet chemistry and laser ablation and analyzed with ICP-MS. Pellets (5 mm diameter) obtained from 40 mg of powder under a pressure of ~1000 MPa (20000 N) were embedded in epoxy resin. The IF-G sample that could not be pelletized due to the heterogeneous size of the minerals forming the mixture after grinding. This size heterogeneity can also lead to a nugget effect on LA-ICP-MS analysis.

**2.3 Wet chemistry**

**2.3.1 Sample digestion**

Every step of the sample preparation is conducted in a Class 10000 clean laboratory at Institut Universitaire Européen de la Mer (IUEM, France), using deionized water obtained on a Millipore® Milli-Q system set at resistivity of 18.2 MΩ, and sub-boiled acids. Sample digestions and purification are performed for magnetite and spinel in a 2 mL Savilex® Teflon microbombs. Vials are pre-washed using a sequence of purified nitric acid ($HNO_3$) and a fluoridric acid (HF)-nitric acid ($HNO_3$)-perchloric acid ($HClO_4$) mixtures at 120°C.

About 0.005 to 0.1 g of IF-G, RB, Al_Spl and synthetic nanomagnetite NMA and NMB powder samples are dissolved with ~10 µL of a 4.49 and 3.67 ng/g in-house [235]U and [230]Th mixed spike (Gautheron et al., 2021), in 2 mL Savilex® microbombs. For both magnetite and aluminous spinel, we use, a mixture of 1.5 mL aqua regia (1 volume of 10.5 N HCl + 3 volumes of 18N $HNO_3$) + 0.5 mL of 29N HF+ 2 drops of concentrated $HClO_4$. Even though aqua regia is by itself, sufficient to ensure the total dissolution of magnetite (Blackburn et al., 2007), HF was added, in order to get rid of possible silicate inclusions. The purpose of $HClO_4$ (evaporation temperature above 180°C) to inhibit the trapping of rare earth elements in fluoride crystals during evaporation (e.g., Li and Lee, 2006; Inglis et al., 2018; Ilyinichna et al., 2020). For magnetite, the acid digestion at 130 °C takes only a couple of hours while for aluminous spinel, which is a more refractory mineral, the acid digestion at 130 °C takes at least 48 h. Furthermore, in order to increase the pressure inside the vial, Ultem® sockets are added around the lids as proposed by Inglis et al. (2018) for zircon digestion.

After complete dissolution, the remaining solutions are evaporated following two steps: 1) 130 °C until HCl, $HNO_3$ and HF was evaporated; and then 2) 180 °C to evaporate $HClO_4$. At the end we obtained a solid residue, and we added 0.5 mL of 1N $HNO_3$ before closing the microbombs and placing them on a hot plate at 100 °C.

**2.3.2 U and Th purification**

U and Th contained in IF-G sample, were purified (mostly by removing Al and Fe) using 1.5 mL of Eichrom© UTeva B resin ion exchange resin columns, as in Douville et al., (2010). Resins are washed using 10 mL of deionized water and conditioned with 1 mL of 3N $HNO_3$. U and Th are eluted with 6 mL of 3N $HNO_3$, and recovered with 4 mL of 3N HCl (Th) and 4 mL of 1N HCl (U).

**2.3.3 U and Th dilution**

After digestion, 15 IF-G, 17 RB, 14 Al_Spl, 14 NMA and 12 NMB solutions were diluted to reach suitable concentrations of Fe (< 1500 µg/g) for HR-ICP-MS analysis. As U and Th contents in magnetite and spinel natural samples are low (< 500 ng/g; Cooperdock et al., 2016; Cooperdock and Stockli, 2018 and Schwartz et al., 2020), the quantitative determination of U-Th abundances can therefore hardly be led on solution with dilution factors higher than 3000, as it is routinely proposed for silicate rocks/minerals, (e.g., Li and Lee, 2006). However, magnetite is made of > 75 wt.% Fe, and aluminous spinel contain

~25 wt.% Fe and up to ~50% Al. The direct analysis of undiluted magnetite/spinel solutions, loaded with these two elements by ICP-MS, is known to induce strong non-spectroscopic (or "matrix") effects on both introduction and ionization (Koch et al., 2002; Steenstra et al., 2019).

## 2.4 Analytical conditions

### 2.4.1 Analysis of U and Th by wet chemistry

The U and Th analyses for purified or diluted solutions were performed using an ICP-MS Thermo® Element XR at IUEM associated to either a PFA nebulizer connected to a standard quartz cyclonic chamber or a nitrogen-supplied desolvating nebulizer (ESI® apex Q Elemental Scientific) introduction system (Potin et al., 2020; Costa et al., 2020), depending on the required level of sensitivity. Isotope dilution analyzes are made possible by the $^{235}U$ and $^{230}Th$ mixed spike additions operated before the sample digestion. In addition, to the U, Th elements, the Mn content ($^{54}Mn$ isotope) was additionally measured. Acidic conditions in the analysis solution (addition of HF) prevented Th loss. Four procedural blanks were run and the blank levels for these measurements were 13 picograms of U and 47 picograms of Th. Between two analyses, a wash sequence was performed using a mixture of $HNO_3$ + HF during 4 minutes, avoiding inter sample contamination.

### 2.4.2 In situ laser ablation sampling and U and Th analysis

In-situ LA-ICP-MS analyses were performed on pellets-using a Compex Pro102 Coherent Laser Ablation System coupled to an ICP-MS Thermo® Element XR at IUEM (e.g., VanKooten et al., 2019; Kubik et al., 2021) operated in its low resolution. The laser has a directional power maintained at 1200 V, it emits at the wavelength of 193 nm (Ar-F type) and has pulse duration in the nanosecond range. The energy output was set at 20 $J/cm^2$, with a laser frequency of 10 Hz. The spot diameters at 160 μm, were adapted on the U and Th contents of the targeted pellets (RB, NMA, Al_Spl). Gas blanks were systematically checked by running 10 cycles of measurements before igniting the laser. The whole surface of pellets was covered by 10 LA-ICP-MS analyzes, each one of them consisted in 30 measurements. In addition, the Mn content ($^{54}Mn$ isotope) was measured with the wet chemistry method by ICP-MS to be used as an internal standard for the ablated mass. The calibration was originally operated using silicate international glass standards, BHVO-2g (OIB basalte), BIR1g (tholeiitic basalt) and BCR2g (basalte) (Gao et al., 2002), and complemented with the NMA nanomagnetite sample. LA-ICP-MS analytical parameters are summarized in Table 2.

**Table 2.** LA-ICP-MS analytical parameters

| Laser ablation system ICP-MS | Compex Pro102 Coherent Laser Ablation System 193 nm |
|---|---|
| Forward Power voltage | 1200 V |
| Pulse duration | Nanoseconds |
| Laser frequency | 10 Hz |
| Pulse Energy | 20 J/cm² |
| Vector gas | $^{40}Ar$ |
| Beam size | 160 µm |
| Analysis | 10 s gas blank; 60 s of signal |
| Internal standard | $^{55}Mn$ (data from wet chemistry) |
| External standards | BHVO-2g, BIR-1g and BCR-2g* |
| In-house reference material | NMA** |

*See description in main text

**Nanomagnetite A

# 3 Results

## 3.1 Wet chemistry U and Th concentration results

The purification protocol was used only on the IF-G sample and U and Th concentrations results (0.02±0.01 and 0.1±0.08 µg/g respectively) are reported on Table 3 and Figure 2. Whereas the measured U content present values that are similar to previously published estimates, Th concentrations display more scattered results, distinct from those of previous studies (Govindaraju, 1995; Dulski, 2001; Bolhar et al., 2004; Kamber et al., 2004; Guilmette et al., 2009; Parks, 2014; Bolhar et al., 2015; Viehnmann et al., 2016).

The range of the U and Th concentrations obtained by wet chemistry method on diluted solutions are reported in Figure 2 and Table 4. U and Th concentrations range from 0.02±0.01 to 45.62±3.40 µg/g and from 0.04±0.03 to 116.01±12.60 µg/g respectively. One can notice that the U and Th contents measured in this study for the RB sample differ by a factor ~100 from the one obtained by Schwartz et al. (2020) (Table 4). This might be explained by the presence in our RB powder of calcite (4%) and titanite (2%) (Supplementary data, Fig. S2) which can contain significant U and Th. Calcite and titanite might have been present initially as inclusion in the selected RB magnetite grains. Indeed, Schwartz et al. (2020) selected inclusion-free magnetite grains after X-ray tomography inspection of each grain. The RB powder has been further used as reference in this study for their homogeneous U and Th contents.

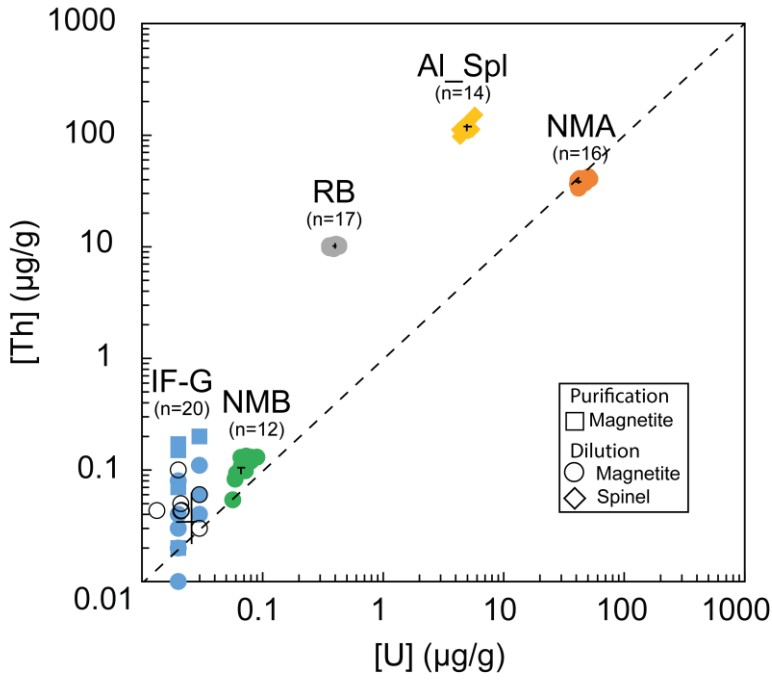

**Figure 2.** U and Th content obtained by wet chemistry and associated mean value and standard deviation (black cross), black empty circles represent IF-G results from the literature (Govindaraju, 1994; Govindaraju, 1995; Dulski, 2001; Bolhar et al., 2004; Kamber et al., 2004; Guilmette et al., 2009; Parks et al., 2014; Bolhar et al., 2015; Viehmann et al., 2016).

Figure 3a represents the dispersion on Th concentrations (expressed as a percentage of these values) as a function of that on U concentrations for each sample. Dispersion magnitude depends on the concentration. The dispersion is larger for Th than for U, whatever the sample. The two samples showing the lower U and Th concentrations, i.e., IF-G and NMB, are those having the most scattered values, 20.2% and 13.8% for U, respectively, and 56.8% and 21.8% for Th, respectively (Figures 3a, b and c). The U and Th content obtained for the IF-G sample is similar to the one obtained in former studies: U concentration ranges from 0.01 to 0.03 µg/g and Th concentration ranges from 0.03 to 0.1 µg/g (Govindaraju, 1995, Dulski, 2001; Bolhar et al., 2004; Kamber et al., 2004; Guilmette et al., 2009; Parks et al., 2014; Bolhar et al., 2015; Viehmann et al., 2016; Fig. 2). The dispersion by 20% and 57% for U and Th, respectively, in IF-G is comparable with dispersion obtained in these literature data. For the RB, Al_Spl and NMA samples, the dispersion of U content is similar and range between 7% and 8% while the dispersion of the Th content is higher for NMA and Al_Spl than for RB even if the latter has a lower concentration (i.e., NMA: 5%; Al_Spl: 11%; and RB: 3%).

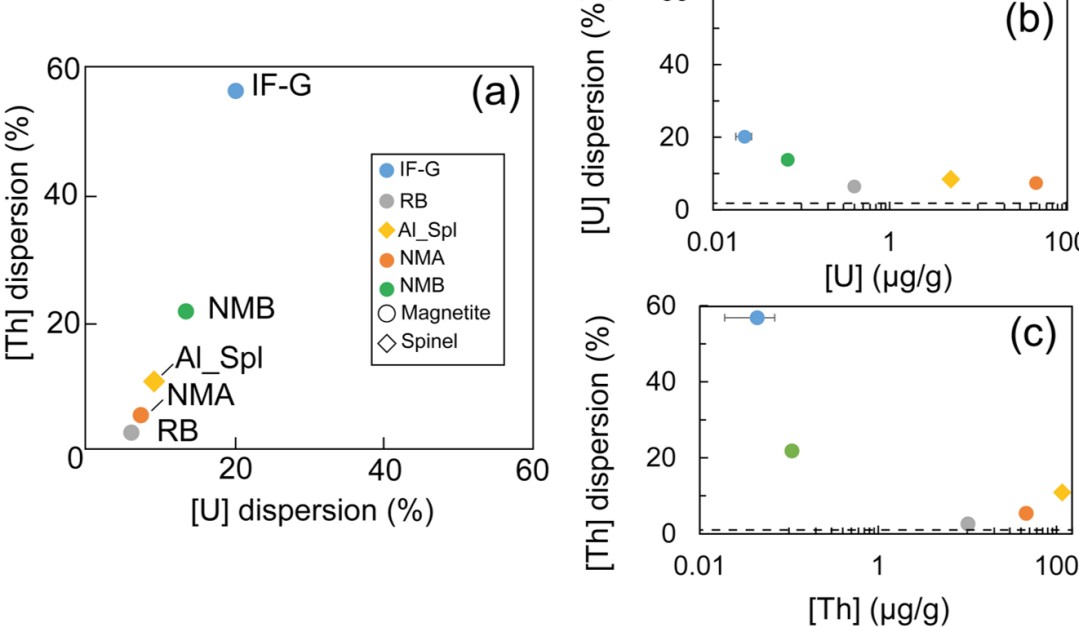

**Figure 3.** (**a**) Dispersion of the U and Th concentration obtained by wet chemistry sampling. (**b**) and (**c**) Mean U and Th concentration as a function of the associated dispersion respectively. Dash lines are the minimum uncertainty of U and Th derived from counting statistics.

**Table 3.** U and Th concentration results obtained by purification.

| Samples | Weight (g) | U (µg/g) | ±2σ | Th (µg/g) | ± 2σ |
|---------|-----------|----------|-----|-----------|------|
| IF-G 16 | 0.0377 | 0.027 | 0.001 | 0.200 | 0.011 |
| IF-G 17 | 0.0450 | 0.024 | 0.001 | 0.166 | 0.017 |
| IF-G 18 | 0.1016 | 0.018 | 0.001 | 0.147 | 0.006 |
| IF-G 19 | 0.0290 | 0.015 | 0.001 | 0.022 | 0.012 |
| IF-G 20 | 0.0381 | 0.025 | 0.005 | 0.075 | 0.014 |
| | | Mean (µg/g) | ± variation coefficient % | Mean (µg/g) | ± variation coefficient % |
| IF-G | | 0.02 | 24.0 | 0.14 | 55.6 |

**Table 4.** U and Th concentration results obtained by dilution.

| Samples | Weight (g) | U (µg/g) | ±2σ | Th (µg/g) | ± 2σ |
|---------|-----------|----------|-----|-----------|------|
| IF-G | | | | | |
| IF-G 1 | 0.0256 | 0.028 | 0.004 | 0.061 | 0.006 |
| IF-G 2 | 0.0076 | 0.031 | 0.004 | 0.058 | 0.014 |
| IF-G 3 | 0.0113 | 0.020 | 0.008 | 0.032 | 0.006 |
| IF-G 4 | 0.0059 | 0.020 | 0.002 | 0.027 | 0.008 |
| IF-G 5 | 0.0061 | 0.019 | 0.004 | 0.035 | 0.014 |
| IF-G 6 | 0.0103 | 0.020 | 0.004 | 0.029 | 0.004 |

| | | Mean (µg/g) | ± variation coefficient % | Mean (µg/g) | ± variation coefficient % |
|---|---|---|---|---|---|
| IF-G 7 | 0.0131 | 0.020 | 0.002 | 0.044 | 0.004 |
| IF-G 8 | 0.0166 | 0.024 | 0.002 | 0.032 | 0.006 |
| IF-G 9 | 0.0034 | 0.018 | 0.002 | 0.038 | 0.006 |
| IF-G 10 | 0.0171 | 0.022 | 0.002 | 0.042 | 0.004 |
| IF-G 11 | 0.0184 | 0.022 | 0.002 | 0.042 | 0.004 |
| IF-G 12 | 0.0129 | 0.021 | 0.002 | 0.077 | 0.006 |
| IF-G 13 | 0.0044 | 0.021 | 0.002 | 0.024 | 0.010 |
| IF-G 14 | 0.0012 | 0.026 | 0.004 | 0.106 | 0.030 |
| IF-G 15 | 0.0026 | 0.020 | 0.004 | 0.011 | 0.001 |
| IF-G | | 0.02 | 20.2 | 0.04 | 56.8 |

| Rocher Blanc | | | | | |
|---|---|---|---|---|---|
| RB 1 | 0.0137 | 0.36 | 0.01 | 10.22 | 0.01 |
| RB 2 | 0.0114 | 0.43 | 0.01 | 10.28 | 0.01 |
| RB 3 | 0.0160 | 0.42 | 0.01 | 10.27 | 0.01 |
| RB 4 | 0.0148 | 0.39 | 0.01 | 9.97 | 0.01 |
| RB 5 | 0.0078 | 0.43 | 0.01 | 10.08 | 0.01 |
| RB 6 | 0.0082 | 0.41 | 0.01 | 10.54 | 0.01 |
| RB 7 | 0.0114 | 0.42 | 0.01 | 10.28 | 0.01 |
| RB 8 | 0.0116 | 0.42 | 0.01 | 10.16 | 0.01 |
| RB 9 | 0.0153 | 0.42 | 0.01 | 10.30 | 0.01 |
| RB 10 | 0.0237 | 0.41 | 0.01 | 10.15 | 0.01 |
| RB 11 | 0.0053 | 0.40 | 0.01 | 10.43 | 0.01 |
| RB 12 | 0.0068 | 0.39 | 0.01 | 10.37 | 0.01 |
| RB 13 | 0.0068 | 0.36 | 0.01 | 9.76 | 0.01 |
| RB 14 | 0.0108 | 0.38 | 0.01 | 10.21 | 0.01 |
| RB 15 | 0.0040 | 0.39 | 0.01 | 9.60 | 0.01 |
| RB 16 | 0.0009 | 0.36 | 0.01 | 9.77 | 0.01 |
| RB 17 | 0.0019 | 0.36 | 0.01 | 9.90 | 0.01 |
| | | Mean (µg/g) | ± variation coefficient % | Mean (µg/g) | ± variation coefficient % |
| RB | | 0.40 | 6.5 | 10.13 | 2.6 |

| Aluminous spinel | | | | | |
|---|---|---|---|---|---|
| Al_Spl 1 | 0.0184 | 5.04 | 0.02 | 109.25 | 0.03 |
| Al_Spl 2 | 0.0236 | 4.93 | 0.05 | 113.81 | 0.02 |
| Al_Spl 3 | 0.0183 | 5.03 | 0.02 | 113.58 | 0.02 |
| Al_Spl 4 | 0.0241 | 5.18 | 0.03 | 109.84 | 0.02 |
| Al_Spl 5 | 0.0133 | 5.19 | 0.01 | 110.26 | 0.01 |
| Al_Spl 6 | 0.0197 | 4.96 | 0.02 | 113.20 | 0.02 |
| Al_Spl 7 | 0.0173 | 5.46 | 0.02 | 112.88 | 0.01 |
| Al_Spl 8 | 0.0204 | 4.36 | 0.02 | 97.11 | 0.01 |
| Al_Spl 9 | 0.0068 | 4.89 | 0.01 | 116.70 | 0.01 |
| Al_Spl 10 | 0.0105 | 5.77 | 0.01 | 151.96 | 0.01 |
| Al_Spl 11 | 0.0028 | 4.50 | 0.01 | 116.44 | 0.01 |
| Al_Spl 12 | 0.0049 | 4.49 | 0.01 | 115.54 | 0.01 |
| Al_Spl 13 | 0.0081 | 4.30 | 0.01 | 111.78 | 0.01 |
| Al_Spl 14 | 0.0034 | 5.08 | 0.01 | 131.84 | 0.01 |
| | | Mean (µg/g) | ± variation coefficient % | Mean (µg/g) | ± variation coefficient % |
| Al_Spl | | 4.9 | 8.4 | 116.0 | 10.9 |

| Nanomagnetite A | | | | | |
|---|---|---|---|---|---|
| NMA 1 | 0.0029 | 50.52 | 0.02 | 42.26 | 0.01 |
| NMA 2 | 0.0092 | 49.48 | 0.04 | 39.84 | 0.01 |
| NMA 3 | 0.0067 | 51.98 | 0.03 | 40.39 | 0.01 |

| | | Mean (µg/g) | ± variation coefficient % | Mean (µg/g) | ± variation coefficient % |
|---|---|---|---|---|---|
| NMA 4 | 0.0085 | 47.32 | 0.02 | 37.26 | 0.01 |
| NMA 5 | 0.0050 | 41.87 | 0.02 | 33.49 | 0.01 |
| NMA 6 | 0.0103 | 45.67 | 0.03 | 38.29 | 0.01 |
| NMA 7 | 0.0065 | 48.01 | 0.01 | 39.19 | 0.01 |
| NMA 8 | 0.0099 | 46.95 | 0.04 | 37.90 | 0.01 |
| NMA 9 | 0.0117 | 47.48 | 0.01 | 37.84 | 0.01 |
| NMA 10 | 0.0058 | 42.85 | 0.03 | 37.84 | 0.01 |
| NMA 11 | 0.0067 | 41.06 | 0.04 | 36.38 | 0.01 |
| NMA 12 | 0.0027 | 43.53 | 0.01 | 39.81 | 0.01 |
| NMA 13 | 0.0044 | 45.95 | 0.01 | 40.25 | 0.01 |
| NMA 14 | 0.0017 | 42.71 | 0.01 | 40.60 | 0.01 |
| NMA 15 | 0.0042 | 43.13 | 0.01 | 40.89 | 0.01 |
| NMA 16 | 0.0019 | 41.34 | 0.01 | 39.25 | 0.01 |
| | | Mean (µg/g) | ± variation coefficient % | Mean (µg/g) | ± variation coefficient % |
| NMA | | 45.6 | 7.4 | 38.8 | 5.4 |
| Nanomagnetite B | | | | | |
| NMB_1 | 0.0079 | 0.07 | 0.01 | 0.13 | 0.01 |
| NMB 2 | 0.0025 | 0.06 | 0.01 | 0.09 | 0.01 |
| NMB 3 | 0.0011 | 0.07 | 0.01 | 0.13 | 0.01 |
| NMB 4 | 0.0026 | 0.07 | 0.01 | 0.11 | 0.02 |
| NMB 5 | 0.0030 | 0.07 | 0.01 | 0.11 | 0.01 |
| NMB 6 | 0.0037 | 0.07 | 0.01 | 0.11 | 0.01 |
| NMB 7 | 0.0022 | 0.07 | 0.01 | 0.10 | 0.01 |
| NMB 8 | 0.0069 | 0.06 | 0.01 | 0.05 | 0.02 |
| NMB 9 | 0.0070 | 0.06 | 0.01 | 0.08 | 0.01 |
| NMB 10 | 0.0042 | 0.08 | 0.01 | 0.13 | 0.01 |
| NMB 11 | 0.0043 | 0.08 | 0.01 | 0.12 | 0.01 |
| NMB 12 | 0.0018 | 0.09 | 0.01 | 0.13 | 0.01 |
| | | Mean (µg/g) | ± variation coefficient % | Mean (µg/g) | ± variation coefficient % |
| NMB | | 0.07 | 13.8 | 0.11 | 21.8 |

240

## 3.2. In-situ U and Th concentrations measured in natural magnetite and spinel using two different calibrations

The calibration of the LA-ICP-MS data used for the data reduction of RB and of Al_Spl samples was performed using silicate glass standards (BCR-2g, BHVO-2g, BIR-1g) together also with the NMA in-house reference sample whose U and Th concentrations were verified by the wet chemistry method.

245 To calculate the U and Th concentration from LA-ICP-MS signal, we used the U/Mn and Th/Mn ratios given for silicate glass standards and the mean value obtained using wet chemistry for the NMA sample and the LA-ICP-MS signal $i(^{238}U)/i(^{55}Mn)$ and $i(^{232}Th)/i(^{55}Mn)$ for the silicate glasses and NMA samples. Thus, for NMA sample, we assumed that the mean U and Th concentrations obtained by wet chemistry method were accurate enough to be used. The U/Mn and $i(^{238}U)/i(^{55}Mn)$ data or the Th/Mn and $i(^{232}Th)/i(^{55}Mn)$ data for each silicate glass sample are well aligned as shown on Figure 250 4 (dotted red line). The results for the NMA sample are slightly shifted from the silicate glass calibration lines, and a second calibration was performed using all results (silicate glass and NMA) that reported on Figure 4 (blue line).

Using this calibration, we calculated the U and Th concentrations for the RB sample using silicate glass standards (U=0.5±0.1 and Th=14.7±2.4 µg/g) and using silicate glasses and NMA samples (U=0.4±0.1 and Th=10.4±1.7 µg/g) (Table 5). Similarly, U and Th concentrations in Al_Spl were determined by LA-ICP-MS using silicate glass standards, yielding U
255   and Th concentrations of respectively 4.1±0.5 and 88.8±8.6 µg/g (Table 5). When NMA is added, derived concentrations are 4.0±0.5 for U and 79.2±7.7 µg/g for Th (Table 5). Dispersion whatever the calibration line remains below 16 % which is comparable with trace element concentration dispersion obtained by Dare et al. (2014) with LA-ICP-MS on magnetite with silicate glass standards (RSD < 15 % for concentration ranging between 10 and 100 µg/g).

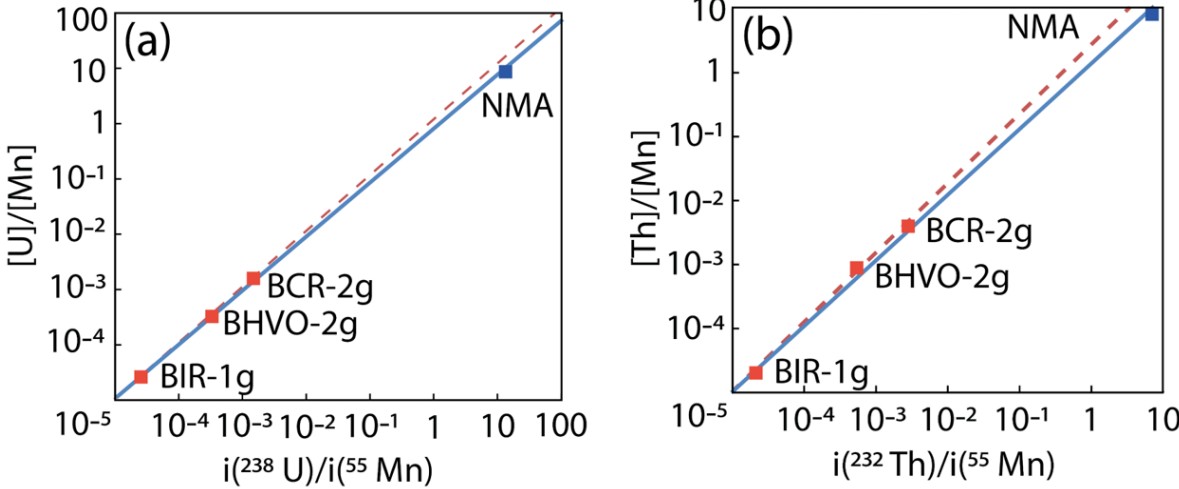

260   **Figure 4.** Calibration lines used for the calibration of the LA-ICP-MS signal. **(a)** U/Mn-calibration lines using silicate glass standards (dotted red line) and NMA nanomagnetite sample (blue line). **(b)** Th/Mn calibration lines with silicate glass standards (dotted red line) and NMA nanomagnetite sample (blue line).

**Table 5.** U and Th concentration determined using LA-ICP-MS method with two types of calibration (silicate glasses only and silicate glasses plus NMA sample).

|  | U (µg/g) | ±1σ | Th (µg/g) | ±1σ |
|---|---|---|---|---|
| Glass standards calibration |  |  |  |  |
| RB 1 | 0.56 | 0.03 | 15.46 | 1.91 |
| RB 2 | 0.45 | 0.03 | 13.08 | 1.91 |
| RB 3 | 0.49 | 0.03 | 12.15 | 1.91 |
| RB 4 | 0.42 | 0.03 | 11.64 | 1.91 |
| RB 5 | 0.39 | 0.03 | 11.60 | 1.91 |
| RB 6 | 0.56 | 0.03 | 14.72 | 1.91 |
| RB 7 | 0.55 | 0.03 | 16.53 | 1.91 |
| RB 8 | 0.55 | 0.03 | 16.53 | 1.91 |
| RB 9 | 0.58 | 0.03 | 16.89 | 1.91 |
| RB 10 | 0.60 | 0.03 | 18.28 | 1.91 |
|  | Mean (µg/g) | ± variation coefficient % | Mean (µg/g) | ± variation coefficient % |
| RB | 0.52 | 14.1 | 14.7 | 16.5 |

| Glass standards and NMA calibration | | | | |
|---|---|---|---|---|
| RB 1 | 0.43 | 0.03 | 10.92 | 1.91 |
| RB 2 | 0.35 | 0.03 | 9.23 | 1.91 |
| RB 3 | 0.38 | 0.03 | 8.57 | 1.91 |
| RB 4 | 0.33 | 0.03 | 8.22 | 1.91 |
| RB 5 | 0.30 | 0.03 | 8.19 | 1.91 |
| RB 6 | 0.44 | 0.03 | 10.39 | 1.91 |
| RB 7 | 0.43 | 0.03 | 11.67 | 1.91 |
| RB 8 | 0.45 | 0.03 | 11.92 | 1.91 |
| RB 9 | 0.45 | 0.03 | 11.92 | 1.91 |
| RB 10 | 0.46 | 0.03 | 12.90 | 1.91 |
| | Mean (µg/g) | ± variation coefficient % | Mean (µg/g) | ± variation coefficient % |
| RB | 0.40 | 14.3 | 10.4 | 16.7 |
| Glass standards calibration | | | | |
| Al_Spl 1 | 4.40 | 0.17 | 95.8 | 2.7 |
| Al_Spl 2 | 3.66 | 0.17 | 86.3 | 2.7 |
| Al_Spl 3 | 4.10 | 0.17 | 89.7 | 2.7 |
| Al_Spl 4 | 3.45 | 0.17 | 71.8 | 2.7 |
| Al_Spl 5 | 5.03 | 0.17 | 104.9 | 2.7 |
| Al_Spl 6 | 3.46 | 0.17 | 87.7 | 2.7 |
| Al_Spl 7 | 4.03 | 0.17 | 81.6 | 2.7 |
| Al_Spl 8 | 4.53 | 0.17 | 92.1 | 2.7 |
| Al_Spl 9 | 4.34 | 0.17 | 89.8 | 2.7 |
| Al_Spl 10 | 3.47 | 0.17 | 88.7 | 2.7 |
| | Mean (µg/g) | ± variation coefficient % | Mean (µg/g) | ± variation coefficient % |
| Al_Spl | 4.05 | 13.3 | 88.8 | 9.7 |
| Glass standards and NMA calibration | | | | |
| Al_Spl 1 | 4.31 | 0.17 | 85.4 | 2.7 |
| Al_Spl 2 | 3.58 | 0.17 | 77.0 | 2.7 |
| Al_Spl 3 | 4.01 | 0.17 | 80.0 | 2.7 |
| Al_Spl 4 | 3.38 | 0.17 | 64.0 | 2.7 |
| Al_Spl 5 | 4.92 | 0.17 | 93.5 | 2.7 |
| Al_Spl 6 | 3.38 | 0.17 | 78.2 | 2.7 |
| Al_Spl 7 | 3.94 | 0.17 | 72.8 | 2.7 |
| Al_Spl 8 | 4.43 | 0.17 | 82.1 | 2.7 |
| Al_Spl 9 | 4.25 | 0.17 | 80.4 | 2.7 |
| Al_Spl 10 | 3.40 | 0.17 | 79.0 | 2.7 |
| | Mean (µg/g) | ± variation coefficient % | Mean (µg/g) | ± variation coefficient % |
| Al_Spl | 3.96 | 13.3 | 79.2 | 9.7 |

## 4 Discussion

### 4.1 Dispersion of wet chemistry U-Th concentration values

The U and Th concentrations measured for the four magnetite (IF-G, RB, NMA and NMB) and the aluminous spinel (Al_Spl) samples using wet chemistry show scattered values with different levels of dispersion (Figs. 2 and 3). The

dispersion of U and Th concentrations could be linked to various parameters. Among them, we identified (i) the influence of the aliquot sample mass, (ii) an over dilution of sample during preparation leading to U-Th concentrations close to the quantification limit or (iii) heterogeneity of the U and Th distribution within the sample. Firstly, the possible impact of the aliquot mass on U and Th dispersion, has been investigated by comparing the dispersion as a function of the sample mass as reported in Figure 5. Although slight dependency of U and Th concentrations on aliquot mass can be observed for some of the samples, statistically there is no obvious correlation between U - Th concentrations and aliquot mass.

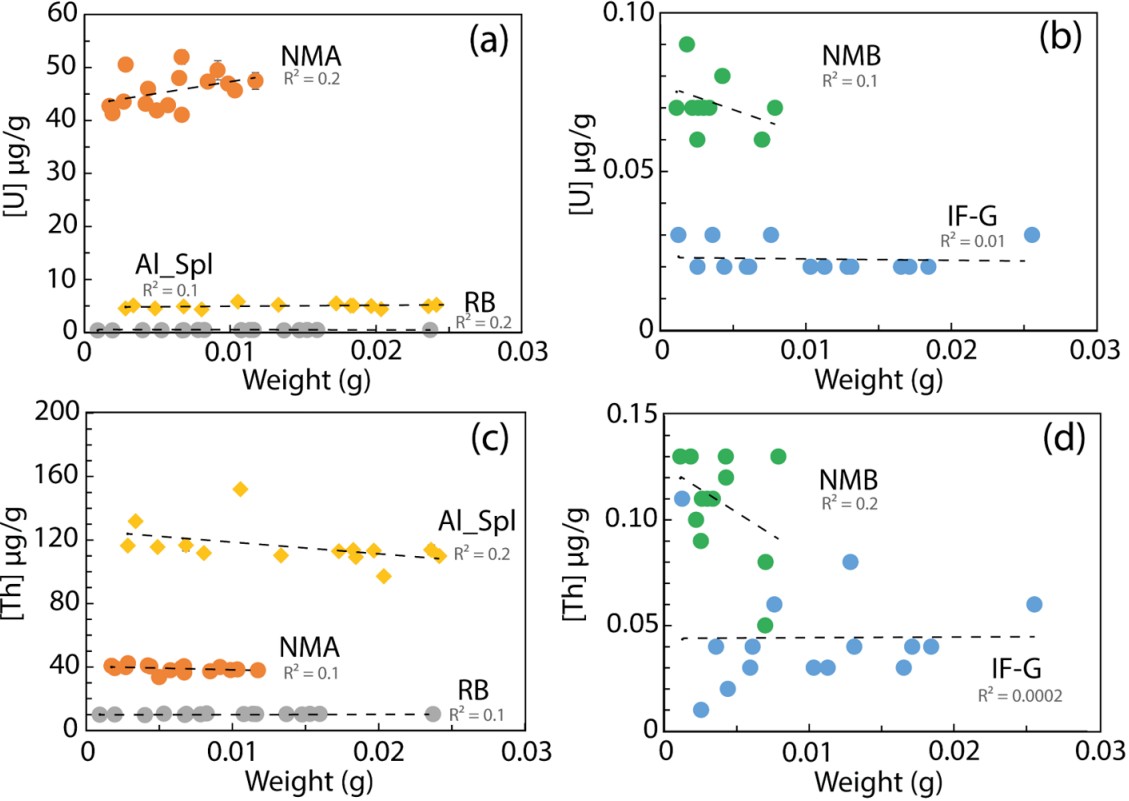

**Figure 5.** U and Th concentration evolution with the mass of dissolved sample. (**a**) U concentration for NMA, Al_Spl and RB samples and (**b**) NMB and IF-G samples, (**c**) Th concentration for NMA, Al_Spl and (**d**) NMB and IF-G samples.

The impact of sample dilution and, thus, the effect of Fe concentration on the ICP-MS plasma were examined. U and Th contents are plotted against the Fe concentration in the analyzed solution on Figure 6. From one sample to another, the Fe content in solution varies from 62 to 1240 µg/g corresponding to a dilution factor comprised between 400 and 5000. The effect of dilution (iron content) on the U and Th analysis precision, if any, cannot account for the dispersion of U and Th concentration. The dilution was sufficient to prevent matrix effect. Moreover, under the analytical conditions indicated in §2.4.1., the limit of quantification was estimated to 0.05 ng/g. This implies that for a sample containing 0.1 µg/g of U and Th (as for samples IF-G and NMB) and for dilution factors higher than 5000, the signal will be close to the quantification limit.

Thus, when measuring magnetite or Al-spinel samples with a low concentration of U and Th, even if the dilution has no statistical effect on the dispersion results, the possibility of reaching concentration below quantification limit should be questioned.

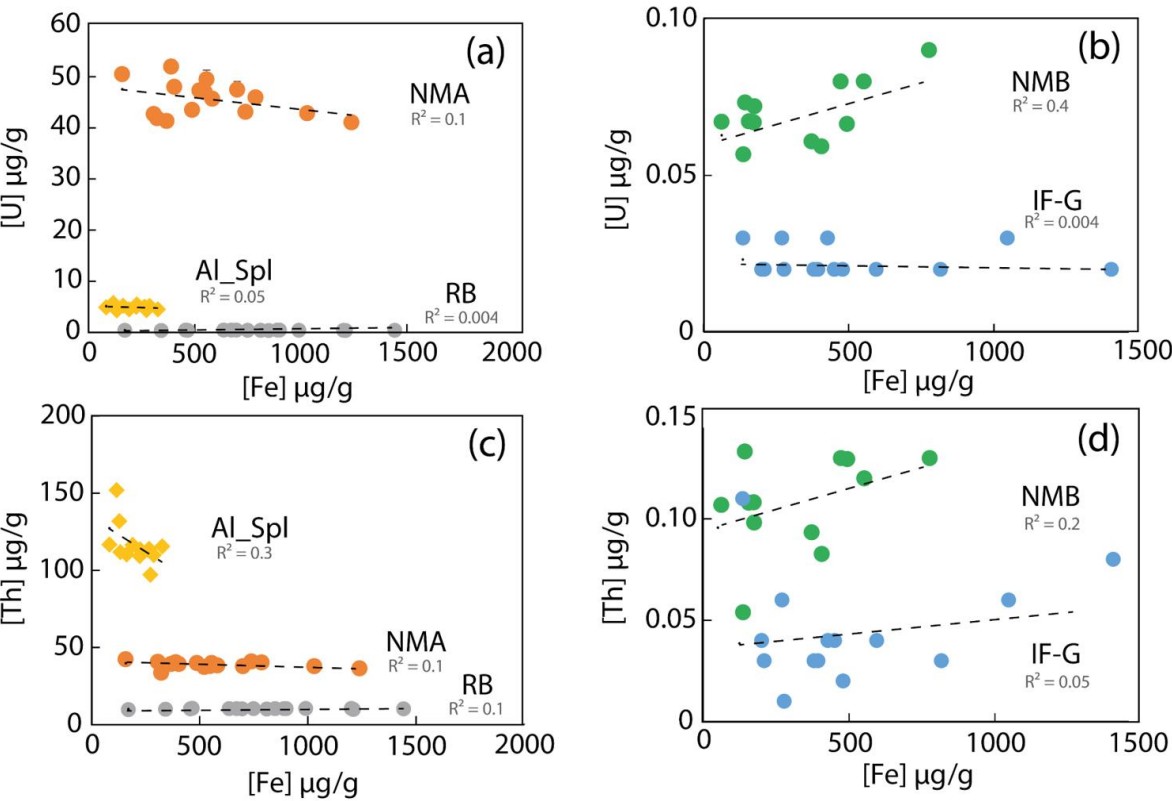

**Figure 6.** U and Th concentrations as a function of iron concentration in the analyzed solution. (**a**) NMA, Al_Spl and RB and (**b**) NMB
and IF-G samples. (**c**) NMA, Al_Spl and (**d**) NMB and IF-G samples.

     Finally, dispersion of U and Th contents could be associated with the chemical heterogeneity of the sample itself. The highest U and Th dispersion is encountered for samples, IF-G and NMB which bear the lowest U and Th contents (< 0.07 and 0.11 µg/g for U and Th respectively). Such a high U and Th dispersion for IF-G sample (20% and 57%, respectively) has been already reported in the literature as shown in Figure 2 and could be associated with the mineralogical heterogeneity of
this sample leading to nugget effect. Nevertheless, NMB sample also yielded high dispersion (14% and 22% for U and Th respectively) compared to NMA sample (7% and 5% for U and Th respectively) even if the two samples have been prepared using exactly the same protocol. The only difference between these samples is the U and Th content. We thus propose that the dispersion of NMB is merely due to concentration effect and the impossibility to measure precisely U and Th content for low value (< 0.1 µg/g) better than 20 %. Finally, despite the identification by XRD of impurities in the RB powder and in the
nanomagnetite samples (NMA), they are the samples for which the U – Th dispersion is the lowest (7% and 5% for U and Th

respectively). For RB, grinding revealed to be efficient at homogenizing mineral phases as well as U and Th over the whole powder. For NMA, the low dispersion of U – Th analyses confirms that nanopowders pressed into pellets are suitable for use as reference material and/or standard (Garbe-Schönberg and Müller, 2014).

**4.2 Accuracy of in situ laser ablation U and Th data**

For magnetite samples, the U and Th concentrations obtained using laser ablation sampling method depends on the reference samples used for calibration (only glass standards or with NMA in addition as external reference material). For the RB sample, the U and Th concentrations obtained using the silicate glass standards are by 30% higher (reference deviation, RD = 30%) than the one obtained with the wet chemistry method, as shown on Figure 7a. However, with the addition of the
NMA sample to the calibration as a reference material, the U and Th concentration show almost identical results (uranium, RD = 0%; thorium, RD = 2%) than the wet chemistry method (U = 0.4±0.03 µg/g and Th = 10.1±0.3 µg/g, Fig. 7b). For comparison, Dare et al. (2014) obtained a RD ~15% for the LA-ICP-MS analyses of various trace elements (0.07 µg/g to 88000 µg/g) in natural magnetite using silicate glasses (NIST-361, MASS-1, BCR-2g) as standards.

A cross-calibration by other laboratories will be the next step to certify the synthetic NMA sample as a standard for U and
Th concentration in magnetite. Furthermore, our synthesis protocol was successful in obtaining nanomagnetite with low U and Th concentrations (NMB: U-Th < 0.13 µg/g). It will be possible to extend the range of U – Th concentration of the synthetic nanomagnetite material if needed.

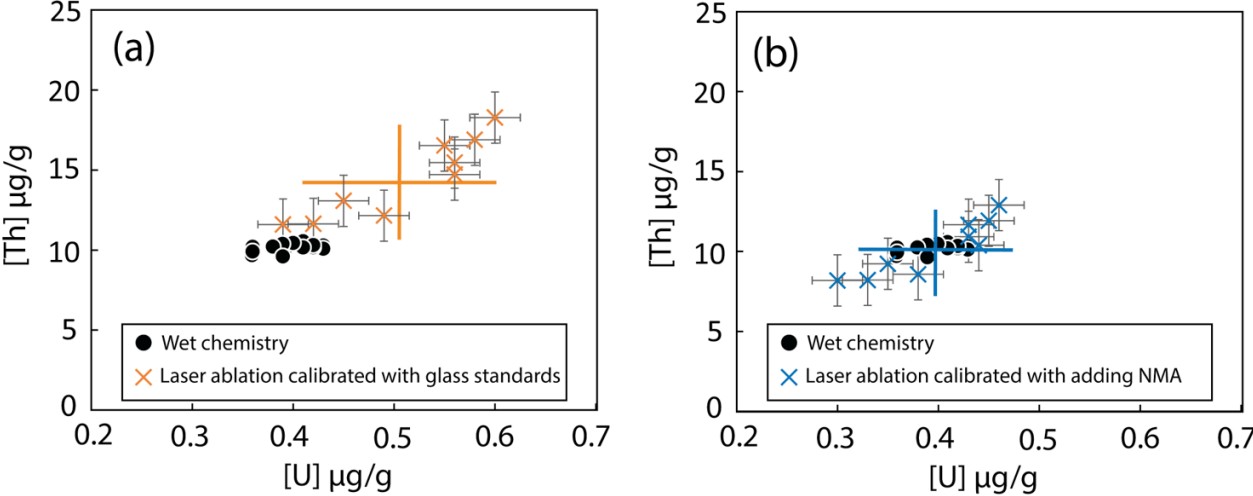

**Figure 7.** Comparison between results obtained by LA-ICP-MS for the two calibrations and by wet chemistry for the RB magnetite sample. **(a)** Calibration with silicate glasses as mean value and dispersion reported with the orange crosses and **(b)** Calibration with addition of NMA sample with mean value and associated dispersion (blue crosses). U and Th contents obtained using wet chemistry protocol are represented by the black dots.

Similarly, for Al_Spl sample, the U and Th concentrations obtained using different standard samples for calibration are
plotted in Figures 8a and b and compared to the wet chemistry results. NMA addition to the calibration has no significant effect on the obtained U which actually differs by more than 19% from the wet chemistry data. (4.0±0.5 µg/g using laser ablation and 4.9±0.4 µg/g using wet chemistry, Figure 8). The Th concentrations obtained with LA-ICP-MS are systematically lower than the concentrations obtained by wet chemistry (116.0±12.6 µg/g) whatever the set of used standards (79.2±7.7 µg/g or 88.8±8.6 µg/g; Fig. 8). We propose that it is likely due to matrix effects, and propose that even if

magnetite and spinel *ss* are from the same structural group (spinel), it is necessary to use a standard that is as close as possible in terms of mineral chemistry to avoid systematic biases on the obtained results. It must be however noted that the precision on U and Th concentration measurement on the Al_Spl sample with LA-ICP-MS is comparable to that achieved with the wet chemistry method. This is very encouraging for the use of LA-ICP-MS to analyze U-Th content of spinel. Provided that appropriate standards are used, it is expected that comparable precision and accuracy can be achieved with the

two methods (laser ablation and wet chemistry), keeping in mind that LA-ICP-MS data are much easier to collect in term of time and cost. The next step is therefore the production of appropriate U and Th Al-spinel standard materials.

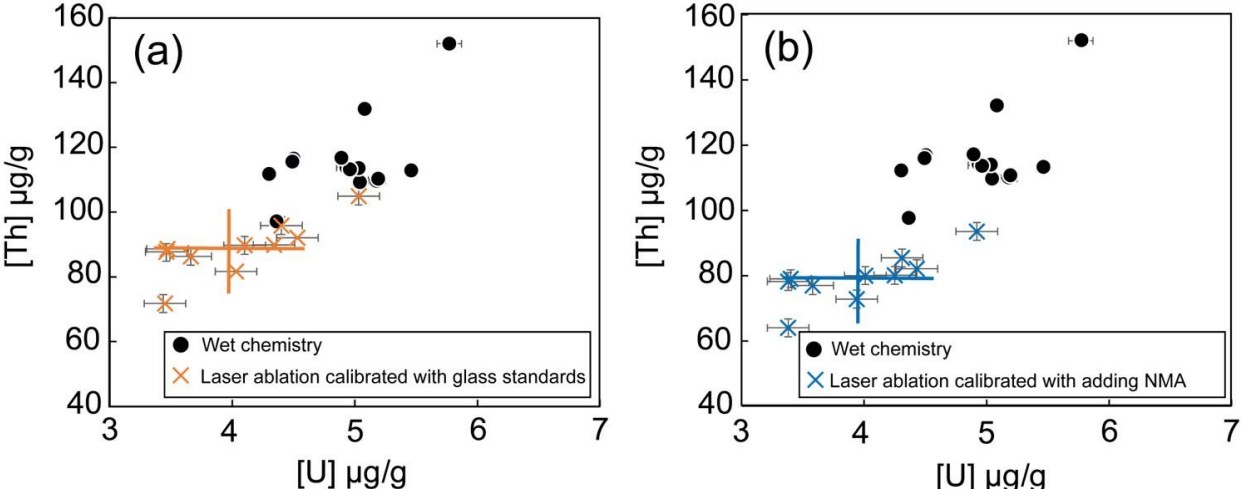

**Figure 8.** U and Th content obtained by wet chemistry and laser ablation method **(a)** Results obtained with silicate glass standards (orange
crosses) and **(b)** with addition of NMA as standard materials (blue crosses). Wet chemistry data are represented by the black dots.

### 4.3 Implication for magnetite and spinel (U-Th)/He thermochronology

In the literature, MgHe dates dispersion ranges from 13 to 70% for crystals containing U and Th below 0.1 µg/g (e.g., Cooperdock and Stockli 2016, Schwartz et al., 2020; Cooperdock et al., 2020), and is less than 5% for magnetite containing U and Th above 0.1 µg/g (Blackburn et al., 2007). Such dispersion includes the analytical errors on the He, U, Th and Sm
measurement. Interestingly, the highest dates dispersion determined on natural magnetite with low U-Th content (< 0.1 µg/g) is higher than the analytical one that can be estimated with the results of this study. For spinel, only one study exists with

SpHe date dispersion of < 9% (Cooperdock and Stockli, 2018). Indeed, dispersion on (U-Th)/He dates includes the analytical errors on He, U and Th concentrations. If the amount of $^4$He is sufficient for a proper analysis with a noble gas mass spectrometer, the analytical error is < 2% (e.g., Gautheron et al., 2021). In this study, we can estimate a dispersion on the

MgHe date of ~24%, for crystals containing U and Th < 0.1 µg/g, and of 4 to 10% for crystals with U and Th content > 0.4 µg/g. We simply propagate the error with a < 2% for the He measurement, and dispersion on U and Th content knowing that they contribute at a different level to the He budget, where U and Th could be combined with the effective uranium (eU) content, with eU =U+0.238×Th+0.0012×Sm (Cooperdock et al., 2019). The dispersion estimated here for MgHe ages, which is based on the analytical errors, is consistent with published dispersions associated with MgHe dating (Cooperdock and

Stockli, 2016; Schwartz et al., 2020; Cooperdock et al., 2020). For Al-spinel, we do not have enough samples to give statistical estimate, but a minimum error of 10% on the SpHe date for the determination of U and Th content > 5 µg/g is anticipated which is commensurable to the uncertainties of spinel ages obtained by Cooperdock and Stockli (2018).

This study confirms that the error on the MgHe and SpHe ages are for a major part due to the difficulty of measuring U and Th concentrations lower than 0.01 µg/g. However, some published MgHe and SpHe data show an age dispersion that is

higher than expected from the present study, which could be associated with alpha implantation from neighbor minerals, mineral inclusions (Schwartz et al., 2020; Hofmann et al., 2021), U-Th zoning, secondary growth of younger magnetite or different He diffusion behavior. This study could not include the effect of heterogeneity due to U or Th mineral zonation which have been erased upon grinding of the samples. However, grinding sample is not best suited to (U-Th)/He geochronology on oxide crystal grain, because of possible loss of material during the crushing. Cooperdock and Stockli

(2016) proposed a protocol to avoid the impact of alpha implantation from neighbor minerals and mineral inclusions by removing the outer crystal shell. Whereas Bassal et al. (2022) showed that He diffusion in magnetite is strongly affected by radiation damage induced by U and Th decay, with typical closure temperature ranging from 200 to 280°C depending on the damage dose and crystal size. As the U and Th content in magnetite crystals from the same geological case present similar values, poor MgHe date dispersion (< 10%) associated with He diffusion changes is expected (Bassal et al., 2022). For

spinel, no quantitative He diffusion coefficient is available limiting the interpretation of the origin of SpHe age dispersion.

In this study, the successful use of LA-ICP-MS with well-suited standards (NMA) opens the possibility to directly access the U and Th distribution across the whole grain for magnetite and spinel. As He content is determined on the bulk grain, if preliminary LA-ICP-MS data show that U and Th are homogeneously distributed within a magnetite grain, then, the MgHe will be estimated with dispersion at 20% if the U and Th content > 0.4 µg/g. More generally, in-situ determination of U and

Th in magnetite or spinel crystals would allow to address the impact of U-Th zoning or secondary growing of hydrothermal magnetite. Finally, compared to wet chemistry methods, acquisition of precise (U-Th)/He dates by LA-ICP-MS may reveal easier, more time efficient, provided that certified and appropriate U - Th standard of magnetite and spinel are used.

## 5. Conclusion

U and Th concentrations ranging from 0.02 to 116 µg/g have been determined in natural magnetite, synthetic U-Th doped magnetite and natural aluminous spinel for the purpose of (U-Th)/He thermochronology. This analytical investigation was based on the comparison of wet chemistry and in-situ laser ablation sampling methods considering their respective advantages and drawbacks. Firstly, we demonstrated that the highest U-Th dispersion is found for the samples with the lowest concentration. This high dispersion shows the difficulty of measuring with HR-ICP-MS, U and Th concentrations below 0.1 µg/g to better than 20%. This implies that magnetite and spinel (U-Th)/He thermochronological dating will yield data dispersion ranging from few percents for U- and/or Th-rich (> 0.4 µg/g) crystals and up to 20% for U-Th poor (< 0.1 µg/g) crystals. Moreover, this study highlights the importance of having new suitable magnetite and spinel reference material to be confident on the analysis U and Th in oxides in wet chemistry and new external standards to ensure accurate analysis of U in-situ laser-ablation sampling. We show that the synthesis of minerals containing U and Th in controlled concentrations can be a way to produce homogeneous and suitable standards for wet chemistry and in-situ analyses.

The use of LA-ICP-MS with synthetic minerals is a promising tool for the acquisition of precise (U-Th)/He dates. Indeed, it allows investigating the distribution of U and Th within a given crystal and evidencing possible growth zones. Dating where U and Th concentration is determined by LA-ICP-MS, however, remains limited by the fact that He is measured on the whole grain or even on multiple grains. Those data can thus be used for dating only if the U-Th distribution is homogeneous in the studied crystals. The synthesis and characterization of U-Th doped-spinel standards is obviously a perspective of this work to access reliable U and Th measurement by wet chemistry and / or laser ablation methods.

## Appendix A: Characterizations of U, Th doped synthetic nanomagnetite

Synthetic nanomagnetite NMA and NMB samples, are enriched in U-Th in the desired concentration range and predicted by our theorical calculations. The synthesis protocol is presented in §2.1.2. The absence of U and Th in the remaining solutions was verified by analyzing these solutions with HR-ICP-MS at IUEM laboratory to ensure that all U and Th are incorporated into the nanomagnetite. In addition, two experiments were done with (i) desorption experiment to quantify the amount of sorbed U and Th sorbed on the surface, and (ii) pH steps dissolution experiment to calculate how much U-Th are sorbed on the surface or incorporated in the structure of the nanomagnetite. These two experiments allow to understand where U and Th are placed: sorbed at surface or present in the structure of nanomagnetite.

## A1 Desorption of U-Th experiments

A first set of U-Th desorption experiments was performed on NMA sample. In a first step, U and Th are desorbed and kept in solution. In a second step, U-Th in solution are complexed in order to measure the complexes with UV-visible. In detail, according to Stopa and Yamaura (2010), U desorption can be achieved using $Na_2CO_3$ and Th can be desorbed using EDTA (Hunter et al., 1988). Two aliquots of NMA (180 mg each) were therefore dipped in respectively 2 mL of 1.1 g/L $Na_2CO_3$

solution and 2 mL of $2.10^{-4}$ mol/L EDTA solution for 40 minutes with shaking. Samples, with U and Th in solution, were

subsequently centrifuged at 13.4 rpm for 15 minutes. Then the concentrations of uranium or thorium in each supernatant were determined by complexing the U and Th by the Arsenazo III method at 650 nm using a UV-Vis spectrophotometer with a detection limit measured at 0.1µg/L (Yamaura et al., 2002).

The UV-Vis result did not detect U-Arsenazo and Th-Arsenazo complexes. According to this experiment there are no U and Th sorbed on the surface of nanomagnetites. The U and Th, seems to be incorporated in the structure of nanoparticles.


### A2 Dissolution of U-enriched-nanomagnetite during pH changes

A different experiment was set up to verify the results obtained by the complexation method (A1). This time, about two grams of nanomagnetite enriched with only 50 µg/g of U were synthesized. After the rinsing step with oxygen-free MilliQ water, the U-doped magnetite was immediately suspended in a reactor containing 300 ml of 0.001 mol/l NaCl solution,

previously deoxygenated for 30 min and kept under N2 bubbling. The suspension was then subjected to decreasing pH steps (8.5, 5.5, 4, 3, 2.3, 1.2 and 1) by addition of 0.1 and 1.5 mol/l HCl. At each step, the pH was kept constant for three hours by automatic addition of HCl using a Titrino Metrohm 716 DMS instrument running Tiamo software. At the end of each pH step, the suspension was sampled, centrifuged, and the supernatant was filtered at 0.20 µm. The Fe and U contents of the different pH-solutions were analyzed by ICP-MS at the Institut de Physique du Globe (France). This allows to follow the

dissolution of uranium and iron according to the pH.

The expected pH of U sorption/desorption was modeled using PHREEQC Version 3 (Parkhurst and Appelo, 2013) using the surface complexation model of Missana et al. (2003) for a specific surface area approximated at 100 m2/g. This pH is, according to this model, is > 4. If U is sorbed on the magnetite surface, solutions of pH 8.5 to 4 are expected to contain U. On the other hand, the expected pH of a magnetite dissolution is less than 4. Thus, if U is contained in the magnetite

structure, solutions at pH 4 to 1 are expected to contain U. Based on the distribution of U in the solutions for each pH step, we can know at the end, the content of sorbed U and the content of U that is in the structure. According to our results, in solutions with pH > 5 there are no U in the solutions. In solutions with pH 5.5 and 4 there is 5% U and 2% Fe. From pH 3 to 1.2 there is 40% U and 20% Fe in solution and at pH 1: 55% U and 75% Fe are in solution. Thus, there is 5% of U sorbed on the surface of nanomagnetite. The majority (95%) of the U is incorporated in the structure of the nanomagnetite. We assume

that Th behaves in a similar way.

**Author contributions.** MC collected and processed data, made figures, and contributed to the writing of the paper. S.S., F.B., A.A., and C.G. initiated the study, processed data, and contributed to the writing of the paper. ML contributed to the synthetize mineral process and contributed to the writing of the paper.

**Competing interests.** The authors declare that they have a possible conflict of interest as Cécile Gautheron is a member of the editorial board of the journal.

**Acknowledgments.** We thank Marie-Laure Rouget for helping to the calibrations of the ICP-MS and the LA-ICP-MS of IUEM. Rosella Pinna-Jamme (GEOPS) and Pierre Burkel (IPGP) are thanked for their help in the measurement of U and Th content analysis of NMA. We thank Nathaniel Findling to helping for the characterization of our samples (SEM, XRD). We thank Bruno Lanson for its help on of the XRD diffractograms interpretation. The associate editor (Daniela Rubatto) and reviewers (Emilie Cooperdock and Florian Hofmann) are warmly thanked for their constructive comments and reviews.

**Financial support.** This research has been supported by the Direction de l'Industrie, des Mines et de l'Environnement de Nouvelle Calédonie (DIMENC).

**Review statement**. This paper was edited by Daniela Rubatto and reviewed by Emily Cooperdock and Florian Hofmann.

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
