# Peer review of "U and Th content in magnetite and Al-spinel obtained by wet chemistry and laser ablation methods: implication for (U-Th)/He thermochronometer"

_EGUsphere, 2022_

## Author Response (AR1)

**Daniela Rubatto (Associate Editor)**

*Comments to the author:*
*Dear Dr Corre*
*I am satisfied with the revision you propose and I invite you to proceed with it as indicated in your reply. Whenever possible address the comments of the reviewer with additions/modifications of the text (I did not see that declared for each answer in the Reply).*

Dear Daniela Rubatto, thank you for your regards on the manuscript and your advices. We add on this file the referee's comments corresponding to the modifications of the new manuscript.

*In adding information on the sample, mineral chemistry (major elements) of the spinel in each sample is also required. If you only have EDS analyses, this may be acceptable, even though fully quantified WDS would be better. Knowing if the samples analysed are Al-Mg or Cr-spinels is rather important. This request that is irrespective of homogeneity of the samples.*

Yes, we only have EDS analyses because for quantifying the major elements, we consider it sufficient.

*The discussion of the potential effect of inclusions in magnetite should definitively be added, if not as a word of caution. Similarly, effects of possible contamination from goethite should be mentioned. I welcome the addition of XRD data to identify inclusions.*

This is indeed a point that we did not raise in the manuscript, as we thought it was irrelevant. We performed XRD analysis for each sample. Actually, the RB powder as well as the nanomagnetites are not purely magnetite. We will discuss this in the text. However, we do not understand what you mean by a goethite contamination since the colloidal synthesis that we have performed precipitates magnetite and goethite at the same time. The goethite is therefore not an external contaminated object but an integral part of the synthesis. Moreover, the U and Th appear homogeneous to the whole powder.

*I agree that the MS can remain as "research article".*

Ok, we will leave the title as it is.

*I recommend that you use the word "standard" only for well characterized (interlab) and well-established materials, otherwise the term "reference material" should be adopted.*

Yes, thank you for this important detail that will improve the quality of our manuscript.

*Line 72. If this is a commonly used standard (or reference material?) then a reference to previous works that describe this material should be added here.*

Yes, it is a commonly used standard and we add now the reference.

*When discussing the LA_ICPMS analysis of magnetite, you may want to consider the work of Dare et al. 2014, https://link.springer.com/article/10.1007/s00126-014-0529-0. They provide details on the methodology in the appendix and a comparison with you approach may be informative.*

Lines 256-258: "Dispersion whatever the calibration line remains below 16 % which is comparable with trace element concentration dispersion obtained by Dare et al. (2014) with LA-ICP-MS on magnetite with silicate glass standards (RSD < 15 % for concentration ranging between 10 and 100 µg/g)."

Lines 308-313: "For the RB sample, the U and Th concentrations obtained using the silicate glass standards are by 30% higher (reference deviation, RD = 30%) than the one obtained with the wet chemistry method, as shown on Figure 7a. However, with the addition of the NMA sample to the calibration as a reference material, the U and Th concentration show almost identical results (uranium,

RD = 0%; thorium, RD = 2%) than the wet chemistry method (U = 0.4±0.03 µg/g and Th = 10.1±0.3 µg/g, Fig. 7b). For comparison, Dare et al. (2014) obtained a RD ~15% for the LA-ICP-MS analyses of various trace elements (0.07 µg/g to 88000 µg/g) in natural magnetite using silicate glasses (NIST-361, MASS-1, BCR-2g) as standards."

*With your revision, please provide a point-by-point account of how you address the criticism and which modifications you made.*

*Kind regards*
*Daniela Rubatto*
*Associate Editor*

**Emily Cooperdock (Referee)**

Referee comment on "U and Th content in magnetite and Al-spinel obtained by wet chemistry and laser ablation methods: implication for (U-Th)/He thermochronometer" by
Marianna Corre et al., EGUsphere, https://doi.org/10.5194/egusphere-2022-520-RC1, 2022

*This paper presents U and Th concentration data and uncertainties for select magnetite and Al-spinel samples with a focus on applicability for (U-Th)/He thermochronology. The primary novel contribution of this study is quantifying the reproducibility between wet chemistry dissolution and laser ICPMS results for samples with different concentration levels of U and Th. In the process of completing the study, they test the impact of matrix matched standards for LA-ICPMS analyses. Overall, this study provides a very helpful and useful scientific contribution on our understanding of the analysis and systematics of U and Th in magnetite and spinel. These are very difficult analyses and the techniques are still in the early stages of becoming more widely applicable. Work like this helps push the method forward and has appeal beyond (U-Th)/He dating (for example, economic geology research is also interested in the trace element chemistry of magnetite and spinel and analytical methods).*
*Overall, I think this manuscript makes an original contribution worthy of publication. Before it is ready to be published, I have several comments, suggestions, and questions for clarification.*

Dear Emily Cooperdock, thank you for your interest concerning the manuscript. Your helpful and constructive comments and suggestions will greatly help improving the manuscript. Please find below the answers to the different points you raised.

*Specific comments:*

*1) More sample information should be provided. These tests were run on 2 natural magnetite samples, 1 natural spinel sample, and 2 magnetite synthetic samples. Magnetite grain habits and inclusions suites can vary significantly between samples. Spinel chemistry can vary significantly as well. It is very possible that different magnetite and spinel samples will have different behaviors in dissolution and/or different analytical challenges in terms of matrix effects and U and Th concentrations. The more these samples are characterized in terms of their crystal habit, age, zonation, inclusion suites and any other known geochemistry, the better for future comparison as more studies include more samples. Table 1 is helpful and 2.1.1 and 2.1.2 have some important background information.*
*Either the main text or the appendix should include more documentation of the sample history and any known geochemical, mineralogy or petrologic characteristics. The study would also benefit from adding photographs of the samples before and after crushing.*

It is true that the maximum of information for each sample should be given, especially with respect to possible matrix effects. It is why we have a table (Table 1) which summarizes the necessary information for all samples. In the next version, we will add photographs of the samples (new Fig. 1) at the relevant scale. XRD data for all samples will also be added in the Appendix.

Lines of the new manuscript 112-113 "Each sample was analyzed by XRD (Fig. S1 to S4). The RB powder contains calcite, titanite, chlorite (Fig. S2), whereas Al_Spl sample is composed of pure aluminous spinel."

*The spinel sample says it is Mg 0.65, Fe 0.35 in Line 84 – how was this determined?*

The structural formula was calculated from spinel compositions determined semi-quantitatively by EDS analysis with a Vega3 Tescan SEM at ISTerre (Grenoble, France). The analyzed samples were homogeneous with respect to major elements. This analytical information will be mentioned in the modified manuscript.

Lines 87-89: "Al-Spl was analyzed by EDS SDD (Silicon Drift Detector) SAMx under a Scanning Electron Microscope (SEM) Vega3 Tescan at the ISTerre (France) and yielded a $(Mg0.65Fe0.35)Al2O4$ composition."

*XRD determined the synthetic magnetite is 85% magnetite and 15% goethite. Why are the XRD results not included in the appendix?*

Effectively, we did not present the XRD results in detail in the previous version to weight down the article, but we agree that the results should be presented and will be in the next version. XRD results of the RB, Al_Spl and IF-G samples in addition to the NMA XRD-pattern will be placed in the Appendix. We will then mention that XRD data were collected in reflective geometry on Bruker diffractometer (D8 Advance at ISTerre (Grenoble, France).

Lines 97-99: "X-ray powder diffraction data diffraction obtained on the NMA sample indicated the production of 85% of magnetite and 15% of goethite and allowed to estimate the grain size to 15 nm from the diffraction peaks width (see Fig S4 in the supplementary data)."

2) *All samples were powdered prior to analysis. Table 1 documents different powdered grain sizes. As far as I know, it is not common to powder aliquots before dissolution during routine (U-Th)/He analysis.*

Effectively, it is not a common routine to powder the samples before digestion and U-Th analysis. We finely powdered the (natural) samples to have homogeneous, and thus identical, samples for both wet chemistry and laser ablation analysis. We will add additional information about the purpose of grinding the samples in the next version.

Lines 363-364: "However, grinding sample is not best suited to (U-Th)/He geochronology on oxide crystal grain, because of possible loss of material during the crushing."

*Is there any evidence that powder grain size impacts U+Th recovery after wet chemistry dissolution? Or was there any observed relationship between powder grain size and laser ablation conditions (pit size, efficiency, matrix effects)?*

We do not have any evidence about the powder grain size and the U-Th recovery, for wet chemistry. However, the grain size might influence the laser ablation conditions (inducing nugget effects), therefore we performed our LA-ICP-MS experiments either on finely grained nanoparticules and glasses and we don't have additional information. This is a question for a future study.

*The Issua sample is a mixture of magnetite, quartz, and actinolite. Does that mean these analyses included a mixture of these minerals or was the magnetite isolated (I assume not based on Table 1)? If it is a mixture, then what is the justification of using the sample to compare with other magnetite? Would such a mixture ever be used for (U-Th)/He analysis?*

We realized that this question was not clearly stated in the original manuscript and will be therefore modified in the next version. The high U and Th dispersion for IF-G sample (20% and 57%, respectively)

has been already reported in the literature and could be associated with the mineralogical heterogeneity of this sample which contains not only magnetite but quartz and actinolite as well. This heterogeneity of the sample makes nugget effects possible and is therefore not a good candidate to be used as a reference sample, for U-Th content and (U-Th)/He date analysis.

Lines 293-295: "Such a high U and Th dispersion for IF-G sample (20% and 57%, respectively) has been already reported in the literature as shown in Figure 2 and could be associated with the mineralogical heterogeneity of this sample leading to nugget effect."

*Is the recommendation of this paper that magnetite and spinel (U-Th)/He should powder samples after degassing and before dissolution? If not, then are the results here translatable to dissolving whole grains? What are the recommendations or warnings to people who may try to do this with whole grains (which is more common for U-Th/He analysis)?*

We will effectively add recommendation in the paper. Our advice would be not to crush the samples before U, Th determination, especially when only small amount of material is available, i.e., some milligrams (1 - 3 mg) and when thus a significant part of the material can be lost upon crushing. With spinel the situation is different since grinding is required to achieve digestion. That is the reason why the use of LA-ICP-MS would be a good alternative to avoid grinding and possible sample loss.

Lines 363-364: "However, grinding sample is not best suited to (U-Th)/He geochronology on oxide crystal grain, because of possible loss of material during the crushing."

*3) A significant portion of the manuscript assesses the potential sources for data dispersion, but there is no discussion of the impact of inclusions or intergrown minerals on the results. One of the known issues with magnetite and spinel (and other opaque phases) is that internal inclusions can be present in unknown quantities and can contribute He, or U-Th-Sm, and/or not be fully dissolved, etc. Prior work tries to get around this by using microCT to screen for and avoid inclusions. Here, some of the samples are reported to include mineral phases other than magnetite (Issua and the synthetic magnetite) determined by XRD.*
*Were the Rocher Blanc magnetite tested for inclusions or intergrown minerals either by microCT or XRD? What about the Al_Spl?*

This is effectively a point that we did not raise in the manuscript, considering that it was beyond its scope. To answer some point raised by the two reviewers about the reason of the RB sample contamination, we performed XRD analyses of the powder and detected the presence of calcite, chlorite, titanite in addition to magnetite. It is true that to acquire a large amount of materials (> 1 g), we grind a lot of grains that likely contain mineral inclusions or other minerals at the edge of the grains. In Schwartz et al., (2020) only clean single grains were selected and analyzed. Here, the few percent of titanite and calcite are very likely responsible for U and Th enrichment found in the RB powder. For the Al_Spl, the sample is pure spinel and no mineral inclusions were observed during SEM characterization.
Additional information will be added in the next version, to better emphasize those points.

Lines 112-113 ". Each sample was characterized by XRD (Fig. S1 to S4). The RB powder contains calcite, titanite, chlorite (Fig. S2), whereas Al_Spl sample is composed of pure aluminous spinel."

*For this study, how could intergrown phases or inclusions impact the dispersion in the data? How would this vary between the wet chemistry technique and LA-ICPMS?*

Indeed, intergrown phases could impact the U and Th dispersion, especially for RB and synthetic nanomagnetite samples that contain titanite-chlorite-calcite and goethite, respectively. However, those samples present quite homogeneous U and Th dispersion for both wet chemistry and laser-ablation. The

integration of these samples during U, Th analyses (during both LA-ICP-MS and wet chemistry) are, in our opinion, adequate to be used as standards.

Lines 117-119: "The IF-G sample that could not be pelletized due to the heterogeneous size of the minerals forming the mixture after grinding. This size heterogeneity can also lead to a nugget effect on LA-ICP-MS analysis."

Lines 299-303: "Finally, despite the identification by XRD of impurities in the RB powder and in the nanomagnetite samples (NMA), they are the samples for which the U – Th dispersion is the lowest (7% and 5% for U and Th respectively). For RB, grinding revealed to be efficient at homogenizing mineral phases as well as U and Th over the whole powder. For NMA, the low dispersion of U – Th analyses confirms that nanopowders pressed into pellets are suitable for use as reference material and/or standard (Garbe-Schönberg and Müller, 2014)"

*Please include a greater discussion on the possibility for these effects within discussion section. A recent study that showed the impact of inclusion in magnetite on He concentration is Hofmann et al., 2021 "Exposure dating of detrital magnetite using 3He enabled by microCT and calibration of the cosmogenic 3He production rate in magnetite" in GChron.*

We will add additional discussion about the effect of inclusion on He budget in magnetite and cite Hofmann et al. (2021).

Lines 358-366: "This study confirms that the error on the MgHe and SpHe ages are for a major part due to the difficulty of measuring U and Th concentrations lower than 0.01 µg/g. However, some published MgHe and SpHe data show an age dispersion that is higher than expected from the present study, which could be associated with alpha implantation from neighbor minerals, mineral inclusions (Schwartz et al., 2020; Hofmann et al., 2021), U-Th zoning, secondary growth of younger magnetite or different He diffusion behavior. This study could not include the effect of heterogeneity due to U or Th mineral zonation which have been erased upon grinding of the samples. However, grinding sample is not best suited to (U-Th)/He geochronology on oxide crystal grain, because of possible loss of material during the crushing. Cooperdock and Stockli (2016) proposed a protocol to avoid the impact of alpha implantation from neighbor minerals and mineral inclusions by removing the outer crystal shell."

*4) Spinel dissolution can be quite challenging. It would be very helpful to include in the appendix the exact procedure used for others to reference and reproduce. The text mentions that some spinel took multiple rounds of acid attack.*
*Did the time it took to dissolve spinel trend with data accuracy or reproducibility? It would be very helpful to know if it impacts U and Th recovery or sample loss. If it doesn't impact the data, that would be very comforting to document. If it does impact the data, it will be important to know. It seems that this study can address this question.*

We will include additional information about the spinel digestion protocol. We dissolved Al-spinel in 2mL of Savillex with 1.5 mL of aqua regia + 0.5 mL of 29N HF + 2 drops of concentrated HClO$_4$. Then we put on top of the Savillex a Savillex Ultem® socket, what increased the pressure inside the vial. The Savillex is then put on a hot plate at 130°C. 48 h are enough to completely dissolve the aluminous spinel. HCl, HNO$_3$ and HF were evaporated at 130°. Finally, HClO$_4$ was evaporated at 180°C.

Lines 133-138: "For magnetite, the acid digestion at 130 °C takes only a couple of hours while for aluminous spinel, which is a more refractory mineral, the acid digestion at 130 °C takes at least 48 h. Furthermore, in order to increase the pressure inside the vial, Ultem® sockets are added around the lids as proposed by Inglis et al. (2018) for zircon digestion. After complete dissolution, the remaining solutions are evaporated following two steps: 1) 130 °C until HCl, HNO3 and HF was evaporated; and then 2) 180 °C to evaporate HClO4. At the end we obtained a solid residue, and we added 0.5 mL of 1N HNO3 before closing the microbombs and placing them on a hot plate at 100 °C."

*5) Many of these analyses are very low concentration and close to blank level. Blanks are not reported. Please add any blank or standard data to the main tables or appendix. Without blanks it is not possible to assess the measurements (were the blank corrected?) and without knowing the blanks reproducibility, it is not possible to propagate the full uncertainty on the measurements, which is central to the study.*

Indeed, we did not report the blanks, we will add blank level in the next version as supplementary data. Four blanks were run and the blank levels for these measurements are 13 picograms of U and 47 picograms of Th.

Lines 164-165: "Four procedural blanks were run and the blank levels for these measurements are 13 picograms of U and 47 picograms of Th."

*Line Comments (some may be repetitive with the comments above):*

*45: "is very little soluble in minerals" should be corrected to "He is not very soluble in minerals" or "has low solubility in minerals"*
We agree we will correct this sentence in the next version of the manuscript.

Line 45: "Helium has low solubility in minerals"

*53: Sentence starting "In addition, well characterized…" is clunky and should be rewritten.*

We agree and we will rewrite this sentence.

Line 54: "In addition, well-established magnetite"

*75-85 (2.1.1.): I'm left wanting more information on the samples. Please include more details. Also, how was the spinel composition determined? Microprobe?*

XRD data and photographs of each sample will be given in the next version of the manuscript. We will explain that the spinel composition was determined by scanning electron microprobe (EDS).

Lines 87-89: "Al-Spl was analyzed by EDS SDD (Silicon Drift Detector) SAMx under a Scanning Electron Microscope (SEM) Vega3 Tescan at the ISTerre (France) and yielded a $(Mg_{0.65}Fe_{0.35})Al_2O_4$ composition."

Lines 112-113 ". Each sample was characterized by XRD (Fig. S1 to S4). The RB powder contains calcite, titanite, chlorite (Fig. S2), whereas Al_Spl sample is composed of pure aluminous spinel."

*93: The natural samples and synthetic samples have different grains sizes after powdering. Does this difference in grain size make a difference in the analyses?*

We have no evidence of any grain-size effect on U and Th data obtained by wet chemistry (digestion). Whereas RB sample grains are micrometer-sized and NMA grains are nanometer-sized, they give similar U and Th reproducibility, 6.5 and 2.6%, respectively, for RB, and 7.4 and 5.4%, respectively, for NMA. Therefore, small grain sizes do not improve reproducibility

*100: The samples were ground up before dissolution. Is this a requirement for dissolution?*

We ground the samples in order to get ca. 2 grams of homogeneous product to perform both LA-ICP-MS and wet chemistry analyses on the same samples.

Lines 110-111: "To compare the U and Th concentrations results obtained with wet chemistry and laser ablation extraction methods, 2 g of natural samples (RB and Al_Spl) were homogenized." But this is not a requirement for dissolution.

*What mass was dissolved per aliquot?*

The dissolved mass is always less than 0.03 g. The exact mass of each aliquot will be given in the new Table 4 in the next version.

*Is powdering samples reasonable for typical (UTh)/ He analysis or would it need to be modified?*

No, as we will explain in the next version of the article, it is better to avoid grinding the sample before dating because during grinding part of the material can be lost and thus (U-Th)/He age determination may be altered.

Lines 362-364: "However, grinding sample is not best suited to (U-Th)/He geochronology on oxide crystal grain, because of possible loss of material during the crushing."

125-129: *Was there a trend in dissolution steps vs U+Th recovery for spinel? Does it affect the accuracy of the measurement? What are microbombs?*

I am not sure I understand this question since we did not follow a step dissolution. Spinel was dissolved with the same method used for magnetite. The micro-bombs that we used are Savillex with a closing lid (Ultem® sockets).

134: *Spinel can contain variable amounts of Fe, Al, Mg, and Cr beyond what is listed here. The chemistry likely makes a difference in the way it dissolves and potentially could relate*

This is true, our dissolution protocol only applies to aluminous Mg-spinel. We are working on the dissolution of Cr-spinel which is indeed abundant in ultramafic rocks.

134: *"The direct analyze" should be "The direct analysis"*

Yes, we will correct this sentence.

135: *Can these elements be removed via column chemistry? Do your results suggest that is an important step to avoid matrix effects?*

Yes, we can purify these elements using Eichrom© UTeva B that retains U and Th but not Fe and Al. We have performed experiments to separate these elements for the IF-G sample. We will add in the next version of the manuscript the purification protocol and the results obtained using this protocol. However, our results show that the matrix effects for U and Th content is not responsible for the dispersion of U and Th data. The effect of iron concentration on U – Th determination is actually shown in Figure 5, it can be seen that we reached the necessary dilution.

Lines 141-144: "U and Th contained in IF-G sample, were purified (mostly by removing Al and Fe) using 1.5 mL of Eichrom© UTeva B resin ion exchange resin columns, as in Douville et al., (2010). Resins are washed using 10 mL of deionized water and conditioned with 1 mL of 3N HNO3. U and Th are eluted with 6 mL of 3N HNO3, and recovered with 4 mL of 3N HCl (Th) and 4 mL of 1N HCl (U). "

Lines 203-207: "The purification protocol was used only on the IF-G sample and U and Th concentrations results (0.02±0.01 and 0.1±0.08 µg/g respectively) are reported on Table 3 and Figure 2. Whereas the measured U content present values that are similar to previously published estimates, Th concentrations display more scattered results, distinct from those of previous studies (Govindaraju,

1995; Dulski, 2001; Bolhar et al., 2004; Kamber et al., 2004; Guilmette et al., 2009; Parks, 2014; Bolhar et al., 2015; Viehnmann et al., 2016)..”
These results are on the new Table 2.

159-163: *This is a very interesting observation (that the powdered samples are 100x higher in U+Th than measured by Schwartz et al., 2020). You say it could be contaminated with U+Th. Is that during preparation? Or is it possible that powdering grains included a lot of inclusions that Schwartz et al 2020 avoided by CT scanning their grains prior to analysis?*

Yes, we expect some “contamination” from ca. 4 wt.% of calcite and 2% of titanite (XRD data). Actually, contrary to Schwartz et al. (2020) we ground a large amount of sample (ca. 2 g) and mineral inclusion/coating may have been present in the sample. Furthermore, the grains were not checked with CT scan for purity.

Lines 211-215: “This might be explained by the presence in our RB powder of calcite (4%) and titanite (2%) (Supplementary data, Fig. S2) which can contain significant U and Th. Calcite and titanite might have been present initially as inclusion in the selected RB magnetite grains. Indeed, Schwartz et al. (2020) selected inclusion-free magnetite grains after X-ray tomography inspection of each grain. The RB powder has been further used as reference in this study for their homogeneous U and Th contents.”

*How much sample was powdered to produce the homogenous, enriched U+Th in this study?*

2 g of each sample were powdered.

Lines 110-111: “To compare the U and Th concentrations results obtained with wet chemistry and laser ablation extraction methods, 2 g of natural samples (RB and Al_Spl) were homogenized. The materials were ground separately.”

*Was the same powder split and used to make the pellet for LA-ICPMS?*

Yes, that is why we ground large quantities of material (2 g) and synthesized large quantities of nanomagnetite (12 g) in order to be able to analyze them by wet chemistry and laser ablation and then compare the results. We will add more details in the next version of the manuscript to be clearer.

Lines 110-111: “To compare the U and Th concentrations results obtained with wet chemistry and laser ablation extraction methods, 2 g of natural samples (RB and Al_Spl) were homogenized.”

And lines 115-119: “The U and Th contents of these 4 samples were retrieved using wet chemistry and laser ablation and analyzed with ICP-MS. Pellets were obtained from 40 mg of powder using a pressure of ~1000 Mpa (20000 N) into 5 mm diameter pellets and embedded in epoxy resin. The IF-G sample that could not be pelletized due to the heterogeneous size of the minerals forming the mixture after grinding. This size heterogeneity can also lead to a nugget effect on LA-ICP-MS analysis.”

169: *“The dispersion is more important for Th” – do you mean more important or larger?*

Yes, we meant "larger" and we will change it in the next version.

171: *Have you considered that the larger Th dispersion is due to Th falling out of solution? This is often a problem for wet chemistry analyses.*

No, because we tried to minimize the loss of Th in solution by the systematic addition of HF in our beakers which keeps Th in the solution.
Line 161: “Acidic conditions in the analysis solution (addition of HF) prevented Th loss.”

*Alternatively, Th wash out times on ICPMS can take significantly longer than U and other elements. Sometimes Th takes a long time to reach the detector compared to U and other elements. Could either of these issues be a possibility for the Th uncertainty?*

No, because the required washing time has been optimized on Th-bearing samples. Washing between two samples was performed with a mixture of $HNO_3$ + HF during almost 4 min before analysis. This "long" washing sequence allows to remove all the remaining Th in the ICP-MS between each analysis.

Lines 162-163: "Between two analyses, a wash sequence was performed using a mixture of HNO3 + HF during 4 minutes, avoiding inter sample contamination."

Figure 2: *Th dispersion appears to be concentration dependent, but also sample dependent. Your IF-G sample appears to have the highest dispersion and lowest concentrations which makes sense with analytical limits on uncertainties. But it is also a sample with three intergrown minerals. Could some of the dispersion be due to heterogeneous mixtures or nugget effects?*

Indeed, the heterogeneity of the IF-G sample could create a nugget effect on the U and Th results, and the dispersion of the U and Th concentrations is surely partly due to this. We will add further explanation in the article to discuss this option.

Lines 293-295: "Such a high U and Th dispersion for IF-G sample (20% and 57%, respectively) has been already reported in the literature as shown in Figure 2 and could be associated with the mineralogical heterogeneity of this sample leading to nugget effect."

*How much sample was homogenized and how large are the aliquots that were analyzed?*

We homogenized 2 g of RB magnetite grains and 2 g of Al_Spl (single crystal). The weight of the aliquots is always less than 0.03 g and we will specify each mass in Table 4 in the next version of the manuscript.

Line 206: *What does "contrasted values" mean?*

We wanted to mean that the values are scattered.

Figure 4: *I note that the sample weights that I asked about in my previous comments are plotted here. Can sample weights be added to a results table so that the reader can reference it more easily throughout the text?*

We agree. We will add the weight of the aliquots in Table 4 of the next version of the manuscript.

240: *Interesting that the glass standards made the LA-ICPMS RB samples 30% higher than the wet chemistry method, which are already 100x higher than Schwartz et al., 2020. Can you expand more on why this matrix effect causes higher concentrations (rather than lower or dispersed)?*

Yes. As shown in Figure 3 of the manuscript, the slope of the calibration line obtained with glass standards is greater than the slope of the calibration line obtained by adding NMA as a standard. A greater slope will result in apparently higher U and Th concentrations.

We do not not have a clear physical explanation but clearly silicates and oxides react differently to laser ablation and behave differently in the mass spectrometer (see Wilson et al., 2002 published on the *The Royal Society of Chemistry* or Steenstra et al., 2019 published on *the Journal of Analytical Atomic Spectrometry*).

261: *There are other studies that have performed LA-ICPMS on spinel in the literature that should be cited here and can be used to discuss how others have matrix matched their standards or any implications your study has on these prior studies. For example, Colas et al., 2014 "Fingerprints of*

*metamorphism in chromite: New insights from minor and trace elements" in Chemical Geology is one but there are others as well.*

Thank you, we were not aware of this article which is very relevant to the present study. Actually, the source article is Locmelis et al. (2011) which presents the natural spinel standard used by Colas et al. (2014). Locmelis et al. (2011) and Colas et al. (2014) used the sample standard as an "unknown" sample to verify the accuracy and the precision of LA-ICP-MS analysis. They do not use as an external standard in order to avoid matrix effects.

Lines 65-67: "Matrix effect in the case of spinel (Cr-spinel) was only investigated so far for elements with much higher concentration (> 10 µg/g; Locmalis et al., 2011 and Colas et al., 2014) than typical U and Th concentration encountered in spinel (< 0,5 µg/g).."

290-300: *This section of the discussion offers no reference to the impact of inclusions on dispersion in magnetite and spinel. This should be added.*

We will discuss the impact of inclusions on dispersion in the next version. Thanks to grinding, the RB powder is homogeneous in U-Th concentration despite the presence of mineral inclusions. The results of U and Th analyses on RB are very little dispersed compared to other samples and IF-G in particular. As already discussed here, IF-G sample may exhibit nugget effect.

Lines 299-303: "Finally, despite the identification by XRD of impurities in the RB powder and in the nanomagnetite samples (NMA), they are the samples for which the U – Th dispersion is the lowest (7% and 5% for U and Th respectively). For RB, grinding revealed to be efficient at homogenizing mineral phases as well as U and Th over the whole powder. For NMA, the low dispersion of U – Th analyses confirms that nanopowders pressed into pellets are suitable for use as reference material and/or standard (Garbe-Schönberg and Müller, 2014)."

297: *The laser ablation parameters should be reported in a table in the text or appendix. How many spots per sample? Were spots averaged? Were some samples/spot sizes under the detection limits? What was the variation in U and/or Th recovery with spot size?*

All this information will be reported in the next version Table 3. We made 10 spots of 160 µm diameter per sample. The location of the spots was chosen so that all the sample surface was covered in order to verify the homogeneity of the pellets. No sample was below the detection limit and all are presented in Table 3.

310: *Do you think the dispersion (20%) is primarily an analytical limitation or a geologic limitation?*

It is only an analytical limitation in our study because our samples are homogeneous and so we do not consider the geologic part.

356-362: This discussion on the location of U in the synthetic magnetite is super interesting. A big question is whether U can be incorporated into the crystal structure or is adsorbed onto the surfaces and the magnetite grows around it. It can have important impact on dissolution and He production. I wonder if this discussion can be moved into the main text?

We prefer that this part remains in the Appendix in order not to weigh the article down.

**Florian Hofmann (Referee)**

Referee comment on "U and Th content in magnetite and Al-spinel obtained by wet chemistry and laser ablation methods: implication for (U-Th)/He thermochronometer" by Marianna Corre et al., EGUsphere,

*This manuscript explores the measurement of U and Th in magnetite and spinel samples to assess the expected uncertainties with modern analytical techniques at different concentration levels. Magnetite and spinel tend to contain low concentrations of U and Th, which makes it difficult to use them for geochronology, but the ubiquity of magnetite and spinel, and the relatively high closure temperate of helium make them interesting target phases. This manuscript presents useful guidance on how to optimize measurements and outlines the limitations of modern analytical techniques. It represents a significant step towards exploring the potential of magnetite and spinel (U-Th)/He dating and making it possible to use these techniques routinely.*

Dear Florian Hofmann, thank you for your interest and we appreciated your constructive and helpful suggestions. Please find below the answers to the different points that you raised.

*In my opinion, this manuscript is well-written, and the data is presented effectively, but it could benefit from minor revisions, as outlined below. The topic fits within the scope of Geochronology, but I would suggest changing the article type to "Technical Note" since it fits the description of that category more than that of a "Research arti*cle".

As we do not clearly propose an analytical protocol but rather discuss how dispersion in U-Th measurements arises and fluctuates for the two analytical methods (wet chemistry and LA-ICP-MS), we do not believe that this manuscript can be considered as a true "Technical Note".

*I agree with the comments by RC1, and I will only mention additional points below:*

*Line 42: Change "fault" to "faulting".*

Thank you, we will correct this sentence in the next version of the manuscript.

*Line 45: Change "radiogenic" to "radioactive".*

Ok, we will correct tit.

*Line 46: Change "neighbor" to "neighboring".*

Yes, we will correct it.

Line 56: *The quoted number of "0.0012%" is incorrect; 0.0012 is the fraction (not percentage) of the contribution of Sm to the effective U concentration (eU), which equates to 0.12%. The exact contribution of Sm to the radiogenic budget depends on the sample dependent U/Sm and Th/Sm ratios. I have seen some samples with low U and Th concentrations and relatively high Sm concentrations in which Sm did contribute to the total amount of 4He to a level above that of the measurement uncertainty. For most samples, the contribution of Sm to the measured amount of 4He is negligible, but can't be ruled out a priori. I agree with not discussing Sm in the manuscript, but the reasons should be clarified.*

Thank you, we will correct this error in the next version of the manuscript. We did not work on Sm because the used spikes were not adapted in term of [149]Sm content, but we agree that Sm should be measured routinely.

Lines 56-57: "Precision in Sm content determination will not be discussed here because data were not acquired in this study, as Sm is routinely analyzed."

Table 1: *Pictures of the samples in addition to the descriptions would be helpful*
.

We agree and we will add photos of each sample before crushing (now in Figure 1).

Line 93: *Was the goethite removed before analyzing the magnetite?*

No, we did not manage to design a goethite separation protocol from magnetite.

Line 99: "The two mineral phases could not be separated."

*Were U and Th partitioned between the magnetite and goethite?*

It is an interesting question, but we do not know precisely the answer. The step-by-step pH decrease experiments presented in Appendix A, suggest that U (and Th) is not adsorbed onto the magnetite / goethite surface. Only synchrotron XANES experiments will allow to address this question and to determine the actual speciation and location of U and Th atoms. Our goal here was to synthesize a product having a composition/crystal-structure the closest to that of magnetite. The standardization results with this product are actually very promising.

*Please discuss this possibility and mention any data you might have.*

In addition to the fact that we lack relevant data to discuss this issue, owing to the good results obtained with our synthetic standard, discussing the U-Th partitioning between goethite and magnetite seems slightly out of scope.

Line 100: *Why were the samples ground into a powder?*

We finely powdered the (natural) samples to have homogeneous, and thus identical, samples for both wet chemistry and laser ablation analysis. We will add additional information about the purpose of grinding the samples in the next version.

Lines 110-112: "To compare the U and Th concentrations results obtained with wet chemistry and laser ablation extraction methods, 2 g of natural samples (RB and Al_Spl) were homogenized. The materials were ground separately in a planetary mill at 500 rpm for 10 min to obtain a grain size of ~2 microns."

*The stated goal is to assess uncertainties as a result of single-aliquot dating of magnetite grains, but this process homogenizes the sample similarly to a two-aliquot approach. As a result, all intra-sample variability is homogenized, which could be due to a U-Th zonation, or could be true age heterogeneity. If the sample has true age zonation, homogenizing the material would result in a meaningless average age. Therefore, this is very different to the single-aliquot approach usually employed for these types of samples. Please discuss what differences can exist and how the results will be relevant for a single-aliquot approach.*

Our purpose here was to prepare chemically homogeneous powders in order to discuss analytical dispersion using two different analytical methods. This study demonstrates that even on a homogeneous powder, significant dispersion on the U and Th concentrations is encountered using the same method. We agree that for natural single grain, additional parameter such as U and Th zoning can affect the (U-Th)/He age. We will add more discussion in the next manuscript version about additional sources of age dispersion.

Lines 345-346: "Interestingly, the highest dates dispersion determined on natural magnetite with low U-Th content (<0.1 µg/g) is higher than the analytical one that can be estimated with the results of this study."

Lines 358-366: "This study confirms that the error on the MgHe and SpHe ages are for a major part due to the difficulty of measuring U and Th concentrations lower than 0.01 µg/g. However, some published

MgHe and SpHe data show an age dispersion that is higher than expected from the present study, which could be associated with alpha implantation from neighbor minerals, mineral inclusions (Schwartz et al., 2020; Hofmann et al., 2021), U-Th zoning, secondary growth of younger magnetite or different He diffusion behavior. This study could not include the effect of heterogeneity due to U or Th mineral zonation which have been erased upon grinding of the samples. However, grinding sample is not best suited to (U-Th)/He geochronology on oxide crystal grain, because of possible loss of material during the crushing. Cooperdock and Stockli (2016) proposed a protocol to avoid the impact of alpha implantation from neighbor minerals and mineral inclusions by removing the outer crystal shell."

Line 132: *I do not understand this sentence: "The quantitative determination of U-Th abundances can therefore hardly be led on too diluted solutions…". Please re-write to clarify.*

By "too diluted" we meant U and Th concentrations reaching the detection limit. This sentence is indeed not clear and it will be rewritten.

Lines 146-150: "As U and Th contents in magnetite and spinel natural samples are low (< 500 ng/g; Cooperdock et al., 2016; Cooperdock and Stockli, 2018 and Schwartz et al., 2020), the quantitative determination of U-Th abundances can therefore hardly be led on solution with dilution factors higher than 3000, as it is routinely proposed for silicate rocks/minerals, (e.g., Li and Lee, 2006)."

Line 135: *Can you matrix-match your standard solutions to counter these matrix effects? Is this an effective strategy or would removing Fe (like suggested by RC1) produce better results? Did you employ this technique here?*

The used standard solutions are in $HNO_3$ with no iron. This could be interesting to compare these results with concentration resulting from matrix-match solution standard calibration.

Yes, we can purify these elements using Eichrom© UTeva B that retains U and Th but not Fe and Al. We have performed experiments to separate these elements for the IF-G sample. We will add in the next version of the manuscript the purification protocol and the results obtained using this protocol. However, our results show that the matrix effects for U and Th content is not responsible for the dispersion of U and Th data. The effect of iron concentration on U – Th determination is actually shown in Figure 5, it can be seen that we reached the necessary dilution.

Lines 141-144 "U and Th contained in IF-G sample, were purified (mostly by removing Al and Fe) using 1.5 mL of Eichrom© UTeva B resin ion exchange resin columns, as in Douville et al., (2010). Resins are washed using 10 mL of deionized water and conditioned with 1 mL of 3N HNO3. U and Th are eluted with 6 mL of 3N HNO3, and recovered with 4 mL of 3N HCl (Th) and 4 mL of 1N HCl (U)."

Lines 201-205: "The purification protocol was used only on the IF-G sample and U and Th concentrations results (0.02±0.01 and 0.1±0.08 µg/g respectively) are reported on Table 3 and Figure 2. The obtained U and Th contents are similar to previously published estimates, (Govindaraju, 1995; Dulski, 2001; Bolhar et al., 2004; Kamber et al., 2004; Guilmette et al., 2009; Parks, 2014; Bolhar et al., 2015; Viehnmann et al., 2016)."
These results are on the new Table 2.

Line 158: *"45.62±3.40" and "116.01±12.60" contain too many significant figures. Uncertainties shouldn't exceed two significant figures, and the measurement should be rounded accordingly.*

Yes, we agree, the two digits is not useful or even meaningful and we will be using 1 digit instead in the modified version.

Line 159: *A possible contamination is very concerning. What are the procedural blank levels for these measurements? How many procedural blanks were run? Do the Figure 1: Add 1:1 line to make deviations more apparent.*

Four procedural blanks were run and the blank levels for these measurements are 13 picograms of U and 47 picograms of Th.

Lines 162-163: "Four procedural blanks were run and the blank levels for these measurements are 13 picograms of U and 47 picograms of Th."

Figure 2*: I'm not sure what the point of breaking the axis is in subfigure (a). There are no values >60% so the axes could just end at 60%.*

Ok, the figure will be modified on the next version of the manuscript.

*For (b) and (c), adding a line for the minimum uncertainty derived from counting statistics would be helpful. This would show the magnitude of other sources of error, e.g., matrix effects. The analytical trends of uncertainties increasing rapidly below 0.5 ppm are in agreement with my own experience working with similar instruments.*

Ok, we will add on these figures the minimum uncertainty derived from counting statistics which are 1.5 % for U and 0.6 % for Th.

Table 2: *There is a mismatch between the number of digits for the measurements and that of the uncertainties.*
*Keep the uncertainties to either one or two significant figures and adjust the rounding of the main values accordingly. Use the same number of significant figures for the mean values and CVs.*

*Give the full sample names and their abbreviations in the table to make it easier to reference.*

Ok, we will add that in the new table 2 (Table 4) of the next version.

*Also, change "Aluminons" to "Aluminous".*

Thank you, we will correct this word.

*The absolute measured amounts of U and Th, as well as the measured (or weighed?) Fe-oxide mass should be given for each sample, along with the results of procedural blanks. This would allow a comparison of the measurement and the blank level/detection limit.*

Indeed, we have not specified this information which are important. The mass of each analyzed aliquot will be added to Table 4. They all have a mass less than 0.03 g. The blank levels for these measurements are 13 picograms of U and 47 picograms of Th.

Section 3.2: *The wording in this section is a bit unclear and should be revised.*

Thank you, we will carefully examine the wording of this section to improve it.
The wording is now: "In-situ U and Th concentration results of naturals magnetite and spinel using two different standardizations"

Line 189: *Change "samples" to "sample".*

Ok, we will correct this in the next version.

Line 191: *Change "those" to "this".*
Yes.

Line 192: *Delete "in mind".*
Ok.

Lines 207-209: *Did you consider the stability of Th in the solution as a possible cause for dispersion? Th is known to be "sticky", and a high level of acidity needs to be maintained to keep it in solution. Typically, this is done with 5-10% HNO3 and/or by adding a small quantity of HF to the solution. Was the dilution done with water or an acid mixture? How was Th stabilized during dilution? Discuss this here and add a detailed description of the dilution procedure to section 2.3.2.*

We do not believe that the dispersion of Th is due to a Th loss from the solution because we ensured that the Th is kept in solution. Indeed, all solutions were diluted with 10 ml of 0.5 N $HNO_3$ and 1 drop of HF was added. We will give this information in the next version of the manuscript.

Line 162: "Acidic conditions in the analysis solution (addition of HF) prevented Th loss."

Line 208: *Explain what you mean by "over dilution". As the solution is diluted, the U and Th count rates are going to diminish, but matrix effects are going to be reduced. Relative to what do you define the "over" dilution?*

The "over dilution" is relative to the blank concentration. If the dilution is too high, although the matrix effect decreases, the concentration of U and Th may be close to the detection limit.

Lines 270-271: "an over dilution of sample during preparation leading to U-Th concentrations close to quantification limit"

Lines 208-209: *The observed natural variability in U and Th concentrations is similar to that of other iron oxides, such as hematite and goethite (see, for example, Hofmann et al., 2020, Chemical Geology). This natural variability, which can be true age inhomogeneity in some samples, highlights the importance of single-aliquot ages that sample small volumes, such as with conventional laser-heating of aliquots in metal packets or laser-ablation.*

Yes indeed, but that is not the purpose of our discussion. We are not saying that it is important to crush the sample when dating. We wanted to quantify what causes U and Th concentration dispersion in order to minimize it. Even if all our samples are homogenized by grinding or by synthesis, it is possible to see that the concentration in U and Th is an important part of the error propagated on the (U-Th)/He age.

Lines 345-346: "Interestingly, the highest dates dispersion determined on natural magnetite with low U-Th content (<0.1 µg/g) is higher than the analytical one that can be estimated with the results of this study."

Lines 358-366: "This study confirms that the error on the MgHe and SpHe ages are for a major part due to the difficulty of measuring U and Th concentrations lower than 0.01 µg/g. However, some published MgHe and SpHe data show an age dispersion that is higher than expected from the present study, which could be associated with alpha implantation from neighbor minerals, mineral inclusions (Schwartz et al., 2020; Hofmann et al., 2021), U-Th zoning, secondary growth of younger magnetite or different He diffusion behavior. This study could not include the effect of heterogeneity due to U or Th mineral zonation which have been erased upon grinding of the samples. However, grinding sample is not best suited to (U-Th)/He geochronology on oxide crystal grain, because of possible loss of material during the crushing. Cooperdock and Stockli (2016) proposed a protocol to avoid the impact of alpha implantation from neighbor minerals and mineral inclusions by removing the outer crystal shell"

Lines 257-258: *Adjust significant figures as above*.

Ok, we will change that in the next version of the manuscript.

Section 4.3: *This is a very helpful section!*

Lines 288-291: *Add references to the relevant literature for these effects.*

Ok, we will add references here.

Line 308: *The hyphenation of "in-situ" is inconsistent throughout the manuscript.*

This is true, we made a choice and used now in-situ.